# A Comprehensive Review of Silver and Gold Nanoparticles as Effective Antibacterial Agents

**DOI:** 10.3390/ph17091134

**Published:** 2024-08-29

**Authors:** Ricardo Aguilar-Garay, Luis F. Lara-Ortiz, Maximiliano Campos-López, Dafne E. Gonzalez-Rodriguez, Margoth M. Gamboa-Lugo, Jorge A. Mendoza-Pérez, Álvaro Anzueto-Ríos, Dulce E. Nicolás-Álvarez

**Affiliations:** 1Clean Technologies, Environmental Process Development and Green Engineering Laboratory, Department of Environmental Systems Engineering, Escuela Nacional de Ciencias Biológicas, Instituto Politécnico Nacional, Mexico City 07738, Mexico; ricardoaguilargaray@gmail.com (R.A.-G.); turtwigh@hotmail.com (M.C.-L.); dgonzalezr1508@alumno.ipn.mx (D.E.G.-R.); jmendozap@ipn.mx (J.A.M.-P.); 2Hormones and Behavior Laboratory, Department of Physiology, Escuela Nacional de Ciencias Biológicas, Instituto Politécnico Nacional, Mexico City 07738, Mexico; luislo27936@gmail.com; 3Faculty of Chemical and Biological Sciences, Universidad Autónoma de Sinaloa, Culiacan 80013, Mexico; margothgamboa.fcqb@uas.edu.mx; 4Bionic Academy, Unidad Profesional Interdisciplinaria en Ingeniería y Tecnologías Avanzadas, Instituto Politécnico Nacional, Mexico City 07340, Mexico; aanzuetor@ipn.mx

**Keywords:** antibiotic resistance, antibacterial properties, Au nanoparticles, Ag nanoparticles, nanotechnology, minimum inhibitory concentration (MIC), minimum bactericidal concentration (MBC), bacterial infections, nanoparticle synthesis

## Abstract

The increasing threat from antibiotic-resistant bacteria has necessitated the development of novel methods to counter bacterial infections. In this context, the application of metallic nanoparticles (NPs), especially gold (Au) and silver (Ag), has emerged as a promising strategy due to their remarkable antibacterial properties. This review examines research published between 2006 and 2023, focusing on leading journals in nanotechnology, materials science, and biomedical research. The primary applications explored are the efficacy of Ag and Au NPs as antibacterial agents, their synthesis methods, morphological properties, and mechanisms of action. An extensive review of the literature on NPs synthesis, morphology, minimum inhibitory concentration (MIC), minimum bactericidal concentration (MBC), and effectiveness against various Gram(+/−) bacteria confirms the antibacterial efficacy of Au and Ag NPs. The synthesis methods and characteristics of NPs, such as size, shape, and surface charge, are crucial in determining their antibacterial activity, as these factors influence their interactions with bacterial cells. Furthermore, this review underscores the urgent necessity of standardizing synthesis techniques, MICs, and reporting protocols to enhance the comparability and reproducibility of future studies. Standardization is essential for ensuring the reliability of research findings and accelerating the clinical application of NP-based antimicrobial approaches. This review aims to propel NP-based antimicrobial strategies by elucidating the properties that enhance the antibacterial activity of Ag and Au NPs. By highlighting their inhibitory effects against various bacterial strains and relatively low cytotoxicity, this work positions Ag and Au NPs as promising materials for developing antibacterial agents, making a significant contribution to global efforts to combat antibiotic-resistant pathogens.

## 1. Introduction

The escalating prevalence of infections due to multidrug-resistant (MDR) bacterial pathogens, against which current antibiotic treatments are less effective as time goes on, poses a significant and growing global health threat. Such antimicrobial resistance exacerbates the morbidity and mortality rates associated with common bacterial diseases [1,2]. Annually, bacterial resistance is linked to 33,000 deaths in Europe alone, underscoring the urgent need for innovative strategies to combat bacteria [3]. The rapid global dissemination of resistance genes demands international cooperation to tackle this escalating public health crisis. In 2024, the World Health Organization (WHO) underscored the urgency of addressing antibiotic resistance by publishing the Bacterial Priority Pathogens List (BPPL), which highlights the bacteria for which new antibiotics are critically needed. This action reflects the growing global concern over antibiotic-resistant bacteria, which pose a significant threat to public health [4]. Furthermore, the WHO emphasized the broader implications of antimicrobial resistance (AMR), stressing that it undermines the effectiveness of modern medicine, complicates treatment protocols, and increases the risk of disease spread, severe illness, and death [5]. To guide research, development, and innovation projects for new antibiotics, the WHO, supported by the University of Tübingen’s Division of Infectious Diseases (Germany), created a priority pathogens list. This list, developed through a multi-criteria decision analysis by international experts, considers factors such as the lethality, transmissibility, and preventability of the infections caused by these pathogens, the extent of existing antibiotic resistance, and the availability of treatment options. The critical priority list, as detailed in Table 1, identifies multidrug-resistant bacteria as significant risks in healthcare settings, particularly when medical professionals use invasive devices. In contrast, the high and medium priority categories include bacteria with increasing resistance rates, often associated with community-acquired infections such as gonorrhea or foodborne *Salmonella* [6]. The rising prevalence of MDR pathogens necessitates alternative treatments, making non-traditional antibacterial agents an area of significant interest to combat antibiotic resistance developed by various pathogenic microorganisms [7]. This urgency has fueled a push towards leveraging the potential of nanosciences, a field intersecting physics, materials science, and biology, to address this challenge [8]. Nanoscience involves manipulating materials at the atomic and molecular scale, and nanotechnology is its practical application, encompassing the observation, measurement, manipulation, assembly, control, and fabrication of material at this scale [9].

The United States National Nanotechnology Initiative defines nanotechnology as a science, engineering, and technology conducted at the nanoscale (1 to 100 nm), characterized by unique phenomena that enable innovative applications across diverse fields, including medicine, engineering, and electronics [10]. This definition outlines two critical aspects of nanotechnology: the control of shape and size at the nanoscale and the exploitation of unique properties emergent at this scale, such as enhanced surface area, quantum effects, and increased reactivity [9]. Recent advances in nanotechnology have enabled the synthesis of novel nanomaterials (NMs) for diverse applications, including industrial and biomedical uses [11]. Among these, NPs defined as NMs with dimensions between 1 and 100 nm are some of the most extensively studied due to their unique properties, such as high surface area (area to volume ratio) [12,13,14]. Metallic NPs have garnered significant attention for their exceptional antibacterial activity, a property known for centuries with metals, metal oxides, and metal salts used historically in treating bacterial infections [15,16,17]. While their use waned with the advent of antibiotics, the rise of MDR has rekindled interest in NPs as potential antimicrobial agents. Among these, Ag NPs have emerged as up-and-coming candidates for treating bacterial infections, though Au NPs are also under intensive investigation for similar applications [18,19]. Although the antibacterial properties of Ag and Au NPs are promising, several problems remain to be solved to exploit their potential in clinical applications fully. One emerging challenge is the development of bacterial resistance to silver nanoparticles. Studies have shown that bacteria can develop resistance to silver, and commercial strains resistant to silver are now available for laboratory tests. This resistance poses a significant threat to the efficacy of silver nanoparticles as antibacterial agents. To counter this, innovative strategies such as the use of Trojan horses, like proteins, are being explored to avoid resistance development [20].

Ag NPs exhibit interactions with various bacterial strains, highlighting their significant potential for antibacterial applications. Researchers have extensively discussed the antibacterial mechanisms of Ag NPs, as shown in (Figure 1), and the precise effect of NPs on bacteria remains to be defined [21]. Two widely accepted antibacterial mechanisms are contact killing and ion-mediated killing. Ag NPs are known to anchor themselves to bacterial cell walls and penetrate them due to their unique physicochemical and biological properties. This penetration can damage the membrane, potentially causing cell content leakage and bacterial death [22]. Variations in cell wall thickness, mainly composed of peptidoglycan, account for the difference in susceptibility between Gram(+/−) bacteria to Ag NPs [23]. The bacterial cell membrane carries a negative charge due to carboxyl, phosphate, and amino acid groups. This feature results in an electrostatic attraction between the positively charged Ag NPs and negatively charged bacterial cell membranes, facilitating the binding and penetration of Ag NPs into the bacteria [22,24].

While the antibacterial mechanisms of Ag NPs have been extensively studied, the mechanisms of internalization of Au NPs still need to be fully understood. This review underscores the importance of understanding the general mechanisms of NP internalization, which include, direct diffusion, and receptor-mediated uptake. These mechanisms are crucial not only in determining the intracellular fate of NPs and their subsequent biological effects but also have significant implications for the field of nanotechnology and biomedicine.

Ag NPs cross microbial cells, interacting with biomolecules like proteins, lipids, and DNA, leading to bacterial dysfunction and death. Specifically, they bind to ribosomes, causing denaturation and inhibiting protein synthesis. Ag NPs have a high affinity for carboxyl and thiol groups in β-galactosidase, which disrupts intracellular functions [26]. Additionally, Ag NPs induce the production of reactive oxygen species (ROS) and free radicals, including hydrogen peroxide and superoxide anion [27,28]. This action results in the inactivation of respiratory chain dehydrogenases and excessive ROS generation, inhibiting cell respiration and growth [29]. Ag NPs also down-regulate antioxidant enzymes, such as glutathione (GSH) and superoxide dismutase, exacerbating ROS accumulation [21]. The ROS surge triggers apoptosis-like responses, lipid peroxidation, GSH depletion, and DNA damage [30]. Wide studies suggest the antibacterial activity of Ag NPs primarily results from the Ag ions released from the NPs [31,32,33]. The surface area of these NPs is a key parameter, with higher surface areas corresponding to a higher concentration of sustainably released Ag, thereby enhancing antibacterial properties [34]. The antimicrobial action of Ag is linked to its interaction with sulfhydryl groups on enzymes and proteins, leading to protein inactivation [35]. Ag interferes with respiratory electron transport and membrane permeability in micro-molar ranges, affecting cellular respiration and energy production [36]. Additionally, Ag could form complexes with nucleic acids, inhibiting cell division, and reproduction [37]. As a heavy metal ion, Ag contributes to increased cellular oxidative stress in microorganisms, representing another antibacterial mechanism [38]. Despite these multiple actions, experts emphasize the synergistic effect between Ag NPs and released Ag ions as the central antibacterial mechanism [39].

Au NPs exhibit antimicrobial effects through multiple mechanisms. They are stable against oxidation in biological media, rendering them non-toxic and biocompatible [40,41]. Au nanoclusters display catalytic activity like several enzymes, promoting increased generation of reactive oxygen species (ROS) that induce oxidative stress in bacteria [42,43]. Additionally, Au NPs are able to bind irreversibly to thiol groups in proteins, such as NADH dehydrogenase, disrupting the bacterial respiratory chain and generating oxidative stress [44]. The interaction of Au NPs with bacterial cells can also lead to membrane disruption, protein denaturation, and DNA damage. These effects are often size-dependent, with smaller NPs exhibiting more significant antibacterial activity due to their larger surface area and higher reactivity.

Physical, green (biological), or chemical methods allow the synthesis of Ag NPs and Au NPs, and each method affects the NP interaction with bacteria differently [45,46]. As detailed in Table 2, physical methods produce high-purity NPs in large quantities, although they often require significant energy, expensive equipment, and specific conditions [41,45]. In contrast, chemical synthesis methods, including electrochemical, sol-gel, and chemical reduction, are more straightforward, scalable, and generally cost-effective; however, they may involve toxic reagents or solvents [47,48]. On the other hand, green synthesis involves organisms or biologically derived reagents, yielding NPs with high solubility, yield, and stability. Despite introducing complexity due to the use of biological components, this method is viewed as a promising approach because of its low potential toxicity and cost-effectiveness, utilizing a wide variety of resources [13,45,49].

The antibacterial activity of metallic NPs (MNPs), such as Ag and Au, is significantly influenced by their synthesis method, size, and shape [10,46,50,51,52,53,54,55,56]. The nanometric size of NPs allows them to infiltrate bacterial cells and influence various cellular processes [57]. The shape of NPs, which can be spherical or non-spherical (e.g., rods, cubic, triangular), influences their antibacterial activity and absorption efficiency. Researchers theorize that the interaction of these NPs with bacterial cell walls varies based on their shape [58,59,60].

The effectiveness of NPs, evaluated through antibacterial assays such as minimum inhibitory concentration (MIC) and minimum bactericidal concentration (MBC), varies based on these morphometric characteristics through synthesis methods [59,61]. These interactions between NPs and bacteria, driven by electrostatic attraction, hydrophobic and receptor-ligand interactions, and Van der Waals forces, are pivotal for designing new antimicrobial agents [60]. This review synthesizes information from various sources on the morphometric characteristics and synthesis methods of Ag and Au NPs and their bacteriostatic (MIC) and bactericidal (MBC) activities, aiming to inform future nanotechnology and microbiology projects and guide the optimal selection of NPs based on the targeted bacterial type.

This research is meticulously organized into several crucial sections. We embark on a comprehensive introduction that outlines the menace of antibiotic-resistant bacteria and the potential of Au and Ag NPs as effective antibacterial agents. The subsequent section, Literature Compilation and Analysis, meticulously details the data extraction and analysis methods. In Data Preprocessing, we thoroughly discuss the steps taken to process the collected data. The section on NP synthesis Methods meticulously explores the different methods—physical, chemical, and green—used to synthesize NPs and compares their implications. NPs Antibacterial Properties thoroughly examines the interaction of NPs with Gram(+/−) bacteria, detailing their mechanisms of action and providing data on minimum inhibitory concentration (MIC) and minimum bactericidal concentration (MBC). Efficacy of NPs Based on Morphology rigorously analyzes how the size and shape of NPs affect their antibacterial activity, with a comparative analysis of Au and Ag NPs. Besides, discrepancies among studies may arise from the notably higher propensity of certain Gram(−) bacteria, such as *E. coli* and *P. aeruginosa*, to develop resistance to Ag NPs following repeated exposures, in contrast to Gram(+) bacteria. Challenges and Need for Standardization meticulously address the variability in testing methods and emphasize the necessity for standardized protocols. Finally, the Discussion addresses the comparison between the cytotoxicity of Ag and Au NPs against bacteria based on morphological characteristics, synthesis methods, and Gram typology. Additionally, the Conclusions, presented with thoroughness, summarize the importance of NPs in combating antibiotic resistance and propose future directions and biomedical applications, instilling a sense of confidence in our professional colleagues.

## 2. Literature Compilation and Analysis

The compilation of studies related to the antibacterial effects of NPs was carried out, and data on the features of the NPs and the exposed bacteria were extracted. A bibliographic search was conducted using keywords and boolean operators across scientific databases, including Elsevier, PubMed, and other sources and search engines like Web of Science (WOS), Scopus, Google Academic, Elicit, and Connected Papers. The focus was on research articles published between 2006 and 2023 on NPs with bacteriostatic and bactericidal effects. It is noteworthy that 94.32 % of the consulted research was published in the last ten years. The inclusion criteria comprised studies that explored the antibacterial effects of Ag or Au NPs. These studies provided meticulous characterizations of the NPs, including their precise shape and size. Furthermore, they reported antibacterial measurements in terms of MIC or MBC, quantitatively expressed in units of micrograms per milliliter (μg/mL).

The process of article selection and analysis is detailed in the flowchart depicted in Figure 2. This flowchart provides a visual summary of the steps taken to compile and analyze the relevant literature.

### Data Features

The searching articles was conducted using a systematic process, ensuring the thoroughness and reliability of the research. This process, depicted in Figure 3, provides a clear roadmap for the research process. The review focused on extracting specific information about the type of nanoparticle (Ag or Au), including their nano-specific descriptors like size and shape and the classification of bacteria according to their cell wall composition Gram (+/−). The method of synthesis, whether physical or chemical, or the information about the company or laboratory if the nanoparticles were acquired from a third party, was also considered. Additionally, this review emphasized the importance of quantitatively documented results of antibacterial measurements, focusing on MIC and MBC values. These values are crucial because studies investigating the effects of Ag and Au nanoparticles on bacteria predominantly report MIC and MBC values, which facilitate clear differentiation between bacteriostatic and bactericidal effects.

## 3. Data Preprocessing

The total of analyzed articles was 89, extracting 458 data points for Ag NPs and 300 for Au NPs. Specifically, the Ag NPs data included 318 MIC and 140 MBC measurements, while the Au NPs data comprised 200 data for MIC and 100 data for MBC measurements. The study explored the antimicrobial activity of Ag and Au NPs against both Gram(+/−) bacteria. The results from various studies showed a range of MIC values from 0.11 to 1200 μg/mL for Ag NPs and 0.00008 to 8000 μg/mLfor Au NPs. The extracted MBC values ranged from 0.22 to 1500 μg/mL for Ag NPs and 0.00008 to 16,000 μg/mL for Au NPs, as summarized in Table 3.

The data presented in Table 3 reflect the reported values from various studies, which encompass a wide range of experimental conditions and bacterial strains. This variation underscores the necessity for standardization of experimental protocols when testing bacterial strains with nanoparticles. Standardized protocols would enable researchers to obtain more precise and comparable MIC and MBC values for specific types of nanoparticles against particular bacterial strains, taking into account critical factors such as nanoparticle size, and shape. Such standardization is essential for enhancing the reproducibility and reliability of research findings in this field.

## 4. NPs Synthesis Methods

### 4.1. Chemical Methods

Lu et al. [51] employed sodium borohydride (NaBH_4_) in their chemical method to synthesize Ag NPs of 5, 15, and 55 nm, achieving MIC values of 6, 12, and 100 μg/mL, respectively, against *E. coli* CCTCC AB 90054. Shekhar et al. [10] also investigated the antibacterial activity of 15 nm Ag NPs synthesized using sodium citrate and NaBH_4_ against *E. coli*. Kvítek et al. [62] utilized ammonia, sodium hydroxide, and D(+) maltose monohydrate to synthesize 26 nm spherical Ag NPs and found a MIC of 1.69 μg/mL against *E. coli* CCM 3955. Despite the differing synthesis methods, including chemical, physical, and green synthesis, and the variety of bacterial strains tested, these studies collectively highlight the relatively consistent antibacterial effectiveness of chemically synthesized spherical Ag NPs against the same bacterial species. The variation in results is attributed to methodological differences and the specific bacterial strains tested in each study.

### 4.2. Physical Methods

NPs synthesis can also be performed by various physical methods, with laser ablation being one of the most significant physical approaches, particularly in synthesizing Ag NPs [45]. The uniformity of NP distribution and lack of solvent contamination in the synthesized thin films benefit physical synthesis methods over chemical ones [46]. Several factors, such as the wavelength of the laser incident on the metal target, the liquid medium, and the period of the laser pulses, determine the quality of the synthesized Ag NPs [63].

A representative example of this method is observed by Perito et al. [61], where they synthesized Ag NPs by pulsed laser ablation in pure water and aqueous lithium chloride solutions, varying the period of the pulses (applying picosecond or nanosecond pulses), obtaining spherical Ag NPs with sizes smaller than 10 nm when tested against *E. coli* XL1Blue and *B. subtilis* 168 found antibacterial activity against both microorganisms for all four NPs formulations. They obtained MIC values ranging from 1.0 to 9.2
μg/mL regarding their bacteriostatic effect. They observed a similar trend with the MBC values, where the antibacterial effect of Ag NPs improved when synthesized in lithium chloride solutions and picosecond pulsing periods in the synthesis of Ag NPs promoted an increase in the antibacterial activity of the NPs against *B. subtilis* but a decrease against *E. coli*. Applying nanosecond pulsing when synthesizing the NPs reversed this effect. The NPs synthesized by this method should be used against a more significant number of bacterial strains to explain the variations between the different formulations of Ag NPs and why this relationship between pulsing periods and antibacterial activity is present since few studies are available on the antibacterial properties of Ag NPs produced by pulsed laser ablation in liquid, even though they show considerably low MICs [64].

### 4.3. Green Methods

Over the past decade, numerous studies have explored the green synthesis of NPs, highlighting its simplicity, cost-effectiveness, environmental friendliness, and scalability for high yields comparable to chemical synthesis methods [16]. Various biological resources, including bacteria, fungi, yeasts, plant extracts, and algae, serve as reducing agents in the biosynthesis of metal NPs, notably Ag and Au NPs [46]. The antibacterial effectiveness of these NPs varies against Gram(+/−) bacteria, influenced by the synthesis organism, components, and extract types used. These factors contribute to the NPs varying morphology, size, and antibacterial properties [65]. Plants, rich in various compounds, including secondary metabolites, can act as both reducing and stabilizing agents, facilitating NP synthesis without generating toxic products [16]. For instance, Ebrahimzadeh et al. [66] synthesized spherical Ag NPs using *Crataegus pentagyna* fruit extract, achieving notable antibacterial effectiveness with a MIC of 0.11
μg/mL and an MBC of 0.44
μg/mL against *E. coli* ATCC 25922. Similar studies by Javan et al. and Singhal et al. [67,68] also utilized plant extracts to synthesize Ag NPs, demonstrating varying antibacterial activity against *E. coli* strains, with MIC values ranging from 16 to 50 μg/mL. These studies collectively illustrate the promising potential of plant-derived green synthesis in producing highly antibacterial Ag NPs. Balakumaran et al. [69] utilized *Bacillus* sp. to synthesize spherical Ag NPs, 8 to 20 nm in size, which exhibited a MIC of 3.125 μg/mL against *E. coli* ATCC 8739. In contrast, Golinska et al. [70] employed acidophilic *Pilimelia collumellifera* strains SL24 and SL19 to produce spherical Ag NPs of approximately 15.9 nm and 12.7 nm. Researchers tested these NPs against *E. coli* ATCC 8739, finding MIC values of 40 μg/mL and 90 μg/mL and an MBC of 120 μg/mL for NPs synthesized by strain SL24.

Hamida et al. [71] used the cyanobacterium *Nostoc * sp. for synthesizing Ag NPs, reported from 8.5 to 26.44 nm, and observed a MIC of 900 μg/mL and an MBC of 1200 μg/mL against *E. coli* ATCC 25922. Despite the similar morphologies of Ag NPs synthesized via green methods, employing different biological reducing agents led to variations in bacteriostatic activity, as reflected in inconsistent MIC values. For *E. coli* strain ATCC 25922, MIC concentrations were 0.11, 16, and 900 μg/mL; for strain ATCC 8739, they were 3.125, 40, and 90 μg/mL. Consequently, the more controlled conditions required for microorganisms, which plant extracts do not need, likely accentuate this variability between the MICs of Ag NPs synthesized from microorganisms and those synthesized from plant extracts [46].

### 4.4. Comparison of Methods

The consulted articles reveal that the bioactive properties of NPs, such as shape and size, are primarily dictated by synthesis conditions, including pH, temperature, pressure, and time [72]. Various chemical compounds of bacteria, fungi, and plant extracts are employed in green synthesizing Ag and Au NPs, as highlighted in different publications [46]. Silver nitrate (AgNO_3_) and chloroauric acid (HAuCl_4_) are the predominant precursors for Ag and Au NP synthesis, respectively, owing to their low cost and chemical stability [73]. Standard fabrication of Au and Ag NPs often involves reducing the salt in the presence of a reducing agent, with a stabilizing agent added to prevent NP aggregation [46]. Sodium citrate is commonly used as a reducing and stabilizing agent in the chemical synthesis of NPs, yielding particles of variable shapes and sizes based on synthesis conditions. For smaller NPs (<10 nm), a potent reducing agent, such as NaBH_4_, is often used to promote rapid nucleation and uniform silver colloids [10].

As shown in Figure 4, the three main synthesis methods for NPs—chemical, physical, and green—each have distinct processes and implications for the properties of the NPs produced. Chemical synthesis often involves the use of reducing and stabilizing agents to control NP size and shape, which can significantly affect their antibacterial activity. Physical synthesis methods, such as laser ablation, produce high-purity NPs without the use of solvents but require significant energy and specialized equipment. On the other hand, green synthesis, using organisms or plant extracts, offers an environmentally friendly and potentially scalable approach, yielding NPs with unique surface properties that can enhance their interaction with bacterial cells. This eco-friendly aspect of green synthesis is particularly relevant in the context of sustainability. Understanding these synthesis methods is crucial for optimizing NPs properties for specific antibacterial applications.

## 5. NPs Antibacterial Properties

### 5.1. Interaction Gram(+/−) Bacteria and NPs

The specific cell wall characteristics of bacteria, which vary significantly between Gram(+/−) strains, influence their susceptibility to metal NPs [74,75]. Studies reveal that Ag NPs effectively eliminate Gram(+/−) bacteria, but 64 % of articles indicate a more substantial effect on Gram(−) strains; this may stem from the structural differences in their cell walls. Gram(−) bacteria have a thin peptidoglycan layer (∼3 nm) covered by an outer cell envelope of lipoproteins, phospholipids, and lipopolysaccharides [76]. In contrast, Gram(+) bacteria possess a rigid peptidoglycan layer thick between 20 and 80 nm, which has fewer anchoring sites for Ag NPs and is more challenging to cross [77]. Ag NPs interact with phosphorus and sulfhydryl groups in extracellular membrane proteins and thiol amino groups in intracellular membrane proteins, disrupting protein structures and cellular functions [78,79]. Gram(−) are more susceptible than Gram(+) bacteria to this interaction, as their thinner peptidoglycan layer offers less protection, and their negatively charged lipopolysaccharides attract positively charged ions released by Ag NPs [65]. This interaction alters the cell wall morphology, increases permeability, and eventually leads to cell death [79].

A quarter of the studies analyzed observed that Gram(+) bacteria were more affected by Ag NPs than Gram(−) bacteria, while 11% reported no difference in effectiveness against both types. Acharya et al. [53] suggest that physical interactions significantly disrupt Gram(+) bacteria cell walls and that the electrostatic attraction between Ag NPs and bacterial surfaces is crucial for antimicrobial activity. Such attraction is noted between negatively charged bacterial cells and positively charged Ag ions. The high content of teichoic and lipoteichoic acids in Gram(+) bacteria, which have a strong negative charge, may sequester free Ag ions [78]. Similarly, studies on the effects of Au NPs are varied: 45% of analyzed articles found Gram(−) bacteria to be more affected, while 31% detected the opposite, and 24% found no significant difference between bacteria types.

Furthermore, variations between studies might stem from the more remarkable ability of some Gram(−) bacteria to develop resistance to Ag NPs after repeated exposures compared to Gram(+) bacteria. The production of flagellin protein from the adhesive flagellum partly contributes to this resistance, promoting Ag NP aggregation [80]. Shekhar et al. [10] observed greater ease of resistance development in Gram(−) than in Gram(+) bacteria. For instance, in their study, *E. coli* strain MTCC 739 (MIC of 60 μg/mL) was less susceptible than *E. coli* MTCC 443 (MIC of 20 μg/mL) but more resistant than *B. subtilis* MTCC 441 (MIC of 30 μg/mL), suggesting a developed resistance mechanism in MTCC 739.

Recent works indicate that Au NPs exhibit different modes of action against Gram(+/−) bacterial cells, including perturbing membrane permeability, altering cellular respiration, inducing oxidative stress, and damaging cell components [57,81]. Bacterial morphology and membrane composition influence the antibacterial effect of Au NPs, affecting the contact area between bacteria and NPs [82,83]. For instance, some articles suggest Au NPs may be more effective against Gram(+) bacteria due to their cell wall structure, which allows easier entry of NPs; this is attributed to the interaction of Au NPs with lysine, a predominant amino acid in Gram(+) bacteria, and the subsequent generation of ROS [84,85]. Conversely, other studies argue Au NPs are more effective against Gram(−) bacteria, owing to their thinner cell walls and the negative charge of their outer membrane, which facilitates electrostatic interactions with positively charged NPs [86,87]. Figure 5 illustrates these mechanisms of action of nanoparticles on the membranes of Gram(+/−) bacteria.

### 5.2. Minimum Inhibitory Concentration (MIC) and Minimum Bactericidal Concentration (MBC)

This review considers variables such as the morphometric characteristics extracted from the articles. These variables, including shape, size, and synthesis method, are crucial in determining the most effective set of features to enhance the antibacterial action of the NPs, depending on the type of bacteria under investigation. The NPs are categorized into spherical shapes (including ovoid and nearly spherical) and non-spherical shapes (including triangular, cubic, irregular, rod, peanut, and hexagonal). NPs are further classified based on their dimensions: particles less than 10 nm are considered small, those in the 10–50 nm range are medium, and those larger than 50 nm are categorized as big.

The analysis has revealed that the synthesis method, shape, and size intrinsically dictate the optimal characteristics of the NPs against bacteria. This knowledge underscores the adaptability of researchers and professionals in the field, as it guides them in prioritizing high antibacterial effectiveness and simplified synthesis. For instance, small spherical Ag NPs obtained through physical synthesis, such as pulsed laser ablation in liquid media, are preferred when targeting Gram(−) bacterial strains. Even when resources or equipment for physical synthesis are lacking, chemical synthesis can be a viable alternative to produce Ag NPs with similar characteristics, showcasing the resourcefulness of the research community.

To make NPs with reduced cytotoxicity compared to Ag NPs, small spherical Au NPs synthesized by chemical methods are a valid option. However, it must be recognized that the antibacterial capacity of these NPs is significantly lower than Ag NPs. To increase the antibacterial effectiveness of Au NPs, synthesizing non-spherical morphologies and medium sizes is an effective strategy, though this approach generally requires more complex synthetic processes than spherical NPs.

## 6. Effect of NPs Based on Morphology

### 6.1. Effect of Size

The goal of controlled metallic NPs synthesis is to build particles with shapes and sizes that are well-suited to interact with specific bacterial cells [57]. Furthermore, the research [88] has established a significant relationship between the size, shape, and concentration of the obtained NPs and their antibacterial properties. According to the extracted data, in a range from 2 to 100 nm, it was found that the antibacterial effectiveness tends to increase as the size decreases; this is because it influences the interaction process between NPs and bacteria since the smaller the size of the NPs, the more likely they are to achieve penetration into the bacterial cells [89]. The article by Shekhar et al. [10] illustrates in detail and systematically the influence of the size of Ag NPs on their antibacterial effect; in this study, they were able to precisely control the nucleation kinetics and growth of the NPs during the synthesis process, by modifying the concentration of the reagents (AgNO_3_, NaBH_4_ and trisodium citrate), volumes, temperature, and pH, obtaining spherical NPs of 5, 7, 10, 15, 20, 30, 50, 63, 85 and 100 nm monodisperse. They evaluated the antibacterial effectiveness of Ag NPs against four bacterial strains, finding that MIC values ranged from 20 to 110, 60 to 160, 30 to 120, and 70 to 200 μg/mL against *E. coli* MTCC 443, *E. coli* MTCC 739, *B. subtilis* MTCC 441 and *S. aureus* NCIM 5021, respectively. Similarly, in the study by Lu et al. [51], they focused on testing how spherical Ag NPs of different sizes affected different bacterial strains, differing from the previous study as they only tested three sizes of NPs, 5 nm (synthesized using AgNO_3_, NaBH_4_, and PVP), 15 nm (synthesized using AgNO_3_, NaBH_4_, PVP and HEPES) and 55 nm (synthesized using AgNO_3_, PEG and PVP). When evaluating the antibacterial activity against six bacterial strains (*A. actinomycetemcomitans*, *F. nucleatum*, *S. sanguis*, *S. mutans*, *S. mitis*, and *E. coli*), they obtained MIC values in ranges of 6 to 50, 12 to 50 and 100 to 200 μg/mL, for Ag NPs of 5, 15 and 55 nm, respectively.

### 6.2. Effect of Shape

The shape is another characteristic that influences the antibacterial efficacy of NPs; regarding Ag NPs, triangular and cubic NPs seemed to have acceptable bacteriostatic effects [50,90]; however, the spherical shape was considerably predominant, from 300 extracted MIC data, 87% were of spherical Ag NPs since it was considered the most suitable for practical applications in colloidal form or the immobilized state [10]. Nevertheless, in the case of Au NPs, the importance of shape variability increases; this is mainly because Au has a lower antibacterial effect than Ag NPs [81,91], so several studies in an attempt to increase their effects modify their characteristics, such as synthesizing non-spherical shapes (rods, peanuts, prisms or hexagonal) or spherical shapes with modifications such as pores in their structures [56,92,93]. However, out of 200 MIC values of Au NPs extracted, only 20.5% corresponded to NPs with non-spherical shapes, but in comparison with non-spherical Ag NPs, Au NPs in the form of peanut, rod, porous spherical, and nanoclusters had higher antibacterial effectiveness than spherical NPs.

Piktel et al. [56] studied this improvement in effectiveness by synthesizing Au NPs of various shapes (rod, star, peanut and porous spherical), using a chemical method consisting of two steps as show in Figure 6.

First, they prepared the Au seed solution with cetrimonium bromide (CTAB), water, HAuCl_4_, and NaBH_4_. The second step of the synthesis consisted of dissolving CTAB in water, adding AgNO_3_, HAuCl_4_, C_6_H_8_O_6_, and the Au seed solution obtained in the first step, and mixing with vigorous stirring at room temperature for 0.5 and 3 h to obtain rod-shaped (average size 45±8 nm along the long axis and 10±3 nm along the transverse axis) and peanut-shaped (average size 60±5 nm along the long axis and 25±5 nm along the transverse axis) Au NPs, respectively. To obtain star shape (cores have a cube shape and a size of 144 nm, as well as 243 nm at the far ends of the arms), they stopped the reaction after 30 min and added a higher concentration of C_6_H_8_O_6_, while porous spherical NPs (sizes around 44±5 nm) they synthesized them at 70 °C. In addition, they synthesized spherical Au NPs for comparison with the other forms, using two different synthesis methods. The first one consisted of extracting the spherical NPs formed in the preparation of the Au seed solution, which they identified as “spherical Au NPs (CTAB)” (average size of 10 nm). In contrast, the second method consisted of heating a trisodium citrate solution to boiling and then adding HAuCl_4_, heating until the color of the solution changed from colorless to pink-red, indicating the formation of spherical Au NPs (average size of 8 nm). After synthesis, they characterized each obtained NPs by scanning transmission electron microscopy (STEM). They evaluated the antibacterial effectiveness of the different forms of Au NPs against different strains of *E. coli*, *P. aeruginosa*, and *S. aureus*, obtaining MIC values in the range of 0.00008–0.04 μg/mL for the rod, peanut, star, and spherical forms with pores, while the spherical ones presented values higher than 20 μg/mL. Since the change in shape shows a significant difference in antibacterial effect, this is likely due to the larger surface area and more protrusions on the surface [56]; this may allow Au NPs to attach bacteria more easily and subsequently rupture the membrane with the protrusions, as well as impede cellular functions such as respiration, inducing cell death [58].

On the other hand, Li et al. [92] synthesized Au nanoclusters with an average size of 2 nm in a single step, using the cationic ligand (11-Mercapto-1-undecanol)-*N*,*N*,*N*-trimethylammonium bromide (MUTAB) as a reducing and stabilizing agent. These nanoclusters showed promising bacteriostatic activity against Gram(+/−) bacteria, presenting MICs from 0.5 to 4 μg/mL for both types. The researchers suggest that the shape-dependent interaction gives the NPs a more remarkable ability to interact with microorganisms. They theorize that inducing local stress on the bacteria’s membrane forms the basis for the antibacterial activity of non-spherical Au NPs [56].

### 6.3. Comparison between Au and Ag

The antibacterial potential of Ag NPs increases as their size decreases from 100 to 20 nm, with the effect being more pronounced for sizes smaller than 10 nm due to favored direct contact with bacteria and electronic effects that enhance NP activity [94]. Similarly, studies on the antimicrobial activity of Au NPs show that their antibacterial properties depend on the size of the NPs, just as observed with Ag NPs. Table 4 highlights the dependence of antibacterial effectiveness on nanoparticle size, shape, synthesis and bacterial strain by comparing the MIC of Ag and Au NPs. The Table 4 presents the minimum inhibitory concentration (MIC) for various strains treated with Ag and Au nanoparticles over a wide range (6–200 μg/mL). Notably, *A. actinomycetemcomitans*, *F. nucleatum*, and *E. coli* MTCC 443 are highlighted for their low concentration and small NP size required to inhibit growth. The table also includes information on the shape and synthesis method of the nanoparticles, providing a comprehensive overview of the factors influencing antibacterial efficacy. However, there is no information available on the total bactericidal concentration (MBC) for these strains when treated with Ag NPs. In contrast, for Au NPs, there are reports of concentrations effective in targeting bacteria, with a particular emphasis on Gram(+) strains, both in terms of bactericidal (MBC) and growth inhibitory (MIC) effects. Lavaee et al. [55] explored this dependence by testing three different sizes of Au NPs (25, 60, and 90 nm), which they obtained from Biometra-T Gradient, Whatman Biometra, Göttingen, Germany. These NPs were characterized and tested on different *Streptococcus* strains (*S. mutans*, *S. sanguinis*, and *S. salivarius*), both standard (ATCC) and patient isolates. The study found MIC values ranging from 0.97 to 3.17 μg/mL (25 nm), 91.61 to 148.21 μg/mL (60 nm), and 232.95 to 500 μg/mL (90 nm), and MBC values ranging from 1.95 to 6.46 μg/mL (25 nm), 125 to 289.28 μg/mL (60 nm), and 217.26 to 1000 μg/mL (90 nm). These findings align with those for Ag NPs, as smaller NPs present a larger surface area, allowing more extensive interaction with bacteria and more accessible nuclear contact [91].

## 7. Challenges and Need for Standardization

### 7.1. Variability in Testing Methods and Techniques

Researchers suggest size, shape, coating, and even the production route of Au NPs affect their antibacterial ability against Gram(+/−) bacteria [10,51,55,57,88,89,91,94,95]. However, regarding shape and size, further research must examine their influence on each type of bacteria. Many studies using Au NPs synthesized by green methods suggest that the positive antibacterial tests are due to the composition of the Au NPs envelope formed during the reduction process, and the stability of the Au NPs [96,97,98]. Green-synthesized Au NPs exhibit antibacterial properties, demonstrating activity either selectively against a single type of Gram(+/−) bacteria or concurrently against both types [57,99]. Therefore, researchers must first conduct further research to elucidate the mechanisms by which NPs exert their antibacterial effects, specifically against the cell surface of Gram(+/−) bacteria. This research is necessary to understand why the effectiveness of NPs, especially Au NPs, tends to vary [95].

In this review, data were used to analyze the variations in MIC and MBC of different Ag and Au NPs, considering synthesis methods, shape, size, and bacterial classification according to the Gram technique. The goal is to identify patterns that could facilitate the selection of NPs in future microbiology and nanotechnology research. However, one of the main limitations of this review is the variability in assays and techniques used to evaluate the antibacterial activity of NPs across different studies. Researchers employ various measurements such as bacterial viability, inhibition zone, minimum inhibitory concentration (MIC), minimum bactericidal concentration (MBC), colony-forming units, and optical density, which are reported using different expressions and metrics. This variability underscores the urgent need for a standardized method to evaluate and report these data more efficiently. In the absence of such a standard, this review focused on extracting data expressed in terms of MIC and MBC, which significantly limited the database. Moreover, the lack of a standardized protocol for bio-assays to test NPs further complicates the comparison and interpretation of results across studies.

### 7.2. Need for Standardized Protocols

The growing body of research on Au and Ag NPs as antibacterial agents underscores the necessity of standardizing synthesis methods and data reporting protocols, particularly for Minimum Inhibitory Concentration (MIC) and Minimum Bactericidal Concentration (MBC) values. The wide range of synthesis methods, NP properties, and reporting practices documented in current literature makes it difficult to compare and replicate results between studies. This variability poses a major challenge for the integration of NPs research with advanced data analysis techniques, such as artificial intelligence (AI), which relies on consistent and quantitative data.

Standardizing synthesis protocols is essential for producing NPs with consistent properties. This involves establishing clear guidelines for the choice of precursor materials, reducing agents, stabilizers, and synthesis conditions. Consistency in these factors will lead to NPs with reproducible size, shape, and surface properties, enabling more reliable comparisons of their antibacterial activities. Moreover, standardized synthesis protocols will facilitate large-scale production and potential clinical applications of NPs, ensuring that the antibacterial properties observed in laboratory settings can be replicated in real-world scenarios.

To achieve standardization, researchers must adopt uniform methodologies for determining MIC and MBC values. This includes using consistent bacterial strains, growth media, incubation times, and methods for measuring bacterial viability. Additionally, MIC and MBC values should be reported in standardized units (e.g., μg/mL) and include detailed descriptions of experimental conditions. Such standardization will not only improve the reliability of individual studies but also enhance the collective understanding of NPs antibacterial properties.

## 8. Importance of NPs against Antibiotic Resistance

NPs synthesis is a complex process influenced by numerous factors, including precursor materials, reducing agents, stabilizers, temperature, pH, and synthesis duration. The absence of standardized protocols results in NPs with varied properties which, in turn, affects their antibacterial efficacy. For instance, chemical synthesis methods can produce NPs with a broad range of sizes and shapes depending on emerging challenges such as resistance development. We will consider incorporating a brief discussion on this topic, highlighting the necessity for innovative strategies, such as the use of Trojan horses like proteins, to mitigate resistance development in future studies and reviews. The specific reagents and conditions used. Physical methods, such as laser ablation, offer high purity but are energy-intensive and require sophisticated equipment. Green synthesis methods, while environmentally friendly, can introduce additional variability due to the biological components involved. Standardizing synthesis protocols is essential for producing NPs with consistent properties. This involves establishing clear guidelines for the choice of precursor materials, reducing agents, stabilizers, and synthesis conditions. Consistency in these factors will lead to NPs with reproducible size, shape, and surface properties, enabling more reliable comparisons of their antibacterial activities. Moreover, standardized synthesis protocols will facilitate the large-scale production and potential clinical applications of NPs, ensuring that the antibacterial properties observed in laboratory settings can be replicated in real-world scenarios.

## 9. Future Directions and Potential Applications

While Ag and Au nanoparticles show potential as antibacterial agents, their complete clinical application is hindered by emerging bacterial resistance. This resistance, already seen in lab tests with commercially available strains, threatens their effectiveness. To combat this, new strategies like using proteins as Trojan horses are being investigated to prevent resistance. These Trojan horses can enhance the delivery and effectiveness of silver nanoparticles, ensuring that they remain a viable option in the fight against antibiotic-resistant bacteria. Hsiao et al. have shown that bacteria can develop resistance to silver, and commercial silver-resistant strains exist for laboratory testing. This resistance poses a threat to the long-term efficacy of silver-based antibacterial treatments. To counteract this, strategies such as the “Trojan horse” mechanism, in which silver is administered intracellularly using carrier molecules such as proteins, have been proposed to prevent the development of resistance [20].

The MIC and MBC are critical metrics for evaluating the antibacterial activity of NPs. However, current studies often report these values using different units and experimental conditions, leading to significant discrepancies in the data. Standardizing the reporting of MIC and MBC values is crucial for enabling comprehensive meta-analyses and integration with AI-based data analysis techniques. Quantitative data are essential for training machine learning models that can predict the efficacy of NPs against various bacterial strains, optimize synthesis parameters, and identify new antibacterial agents.

To achieve standardization, researchers must adopt uniform methodologies for determining MIC and MBC values. This process includes using consistent bacterial strains, growth media, incubation times, and methods for measuring bacterial viability. Additionally, MIC and MBC values should be reported in standardized units (e.g., μg/mL) and include detailed descriptions of experimental conditions. Such standardization will not only improve the reliability of individual studies but also enhance the collective understanding of NP antibacterial properties.

Building on this standardization, the potential of Artificial Intelligence (AI) and machine learning (ML) techniques to revolutionize antibacterial research is immense. These powerful tools can analyze large datasets and identify patterns that may not be apparent through traditional analysis methods. The integration of standardized NPs data with AI can truly transform the field of antibacterial research. For instance, ML algorithms can be trained on standardized MIC and MBC datasets to predict the antibacterial efficacy of NPs with specific characteristics. These models can also identify the most critical factors influencing antibacterial activity, guiding the design of more effective NPs [100].

Furthermore, AI can facilitate the discovery of new antibacterial agents by analyzing patterns in the data and suggesting novel NPs compositions or synthesis methods. This data-driven approach is crucial in accelerating the development of next-generation antibacterial treatments, addressing the urgent need for new solutions in the face of rising antibiotic resistance. Studies have demonstrated the potential of AI in various biological applications, such as oncology and personalized medicine, showcasing its versatility and power in handling complex biological data [101,102]. Recent research has specifically applied AI techniques to address the issue of antibiotic resistance, a critical concern in antibacterial nanoparticle research. By leveraging large datasets, AI models have been able to predict bacterial resistance patterns and identify effective nanoparticle formulations, significantly enhancing the understanding and development of antimicrobial agents [103]. This underscores the importance of high-quality, standardized data, which AI can use to generate actionable insights for combating antibiotic resistance.

Despite the clear benefits of standardizing synthesis methods and data reporting, several challenges must be addressed. First, there is a critical need for agreement within the scientific community on the optimal protocols for NPs synthesis and MIC/MBC determination. Achieving this requires not just individual efforts but collaboration between researchers, standardization bodies, and regulatory agencies to develop and disseminate guidelines. Second, researchers must implement these standardized protocols and reporting practices in their work. Doing so may involve additional training and resources, particularly for laboratories that currently lack the necessary equipment or expertise. Funding agencies and journals can play a crucial role by prioritizing and promoting standardized methodologies in NPs research.

## 10. Patents and Clinical Trials

### 10.1. Patents on Nanoparticles

Nanoparticles, due to their unique properties and biomedical applications in various fields, have been the focus of numerous patents. The development and application of Ag and Au NPs in the biomedical field have led to a significant number of patents aimed at enhancing the synthesis, stability, and antibacterial efficacy of NPs. These patents are a testament to continuous innovation in nanotechnology, playing a crucial role in addressing the challenges of antibiotic resistance and improving clinical outcomes, offering hope for the future of medicine.

One of the strategic importance of patents lies in their ability to standardize synthesis protocols and utilize NPs for antimicrobial testing, ensuring the reproducibility of experiments and the reliability of results. The key areas covered by these patents include synthesis methods, antibacterial applications, and combinatory treatments, among others. This strategic approach not only advances the field of nanotechnology but also provides a robust framework for developing effective nanoparticle-based antibacterial strategies.

Recent advancements in NP synthesis methods are exemplified by the patent “Direct Formation of Gold Nanoparticles Using Ultrasound” (US Patent No. 10,500,643), which details an innovative green chemistry method for synthesizing Au NPs directly from bulk gold sources. This process employs ultrasonication in water with an alkylthiol species and a quaternary ammonium surfactant, eliminating the need for toxic gold dissolution and reduction steps. This technique, applicable to various bulk gold sources, including electronic waste recovery, produces Au NPs with well-defined plasmon resonance, essential for applications in theranostics and sensor technologies [104].

Similarly, patent EP 2 789 235 A1 outlines methods and compositions for preparing antimicrobial Ag NPs, the process involves creating a solution with a stabilizing agent, a silver compound, a reducing agent, and a solvent, treating medical devices like catheters and wound dressings to provide long-lasting antimicrobial properties [105].

The patent “Method for preparing noble metal nanoparticles” (EP 2878401 B1) describes a green chemistry method for synthesizing gold and silver nanoparticles using extracts from *Olea europaea* fruit and *Acacia nilotica*. This environmentally friendly, cost-effective method produces nanoparticles suitable for medical treatments, including antibacterial and cancer therapies, without high temperatures or toxic chemicals [106].

In antibacterial applications, the patent “Controlled Synthesis of Highly Monodispersed Gold Nanoparticles” describes synthesizing gold nanoparticles with highly uniform sizes (30 to 90 nm) using seed nanoparticles and a reducing and capping agent like acrylate. Controlling parameters like concentration, pH, and temperature achieves a size monodispersity with a standard deviation as low as 2%, crucial for applications requiring specific nanoparticle characteristics [107].

Patent EP 1 889 810 A1 discusses improved methods for producing antimicrobial nanoparticle-loaded silica powders using flame spray pyrolysis. Silica particles doped with silver and copper release metal ions upon contact with moisture, enhancing antimicrobial properties. The patent emphasizes controlling the size, distribution, and concentration of dopant nanoparticles to optimize antibacterial efficacy and explores methods for controlling the color and improving dispersion in various substrates [108].

Additionally, patent WO 2022/245578 A1 explores a nanomaterial composed of gold or silver nanoparticles functionalized with extracts from sweet gum leaves, demonstrating significant antiviral and antibacterial efficacy, particularly against SARS-CoV-2 and influenza viruses. Green synthesis methods enhance bioavailability and therapeutic potential [109].

Lastly, patent US 2012/0046482 Al outlines a novel method for synthesizing Au NPs using a gold ion-containing solution mixed with a carboxylic acid with at least two carboxyl groups, serving as both a stabilizing and reducing agent. The process, reacting at temperatures between 20 and 60 °C, produces Au NPs with varied morphologies, including nanoplates, nanonetworks, and nanochains, critical for applications in catalysis, biosensing, and optoelectronics [110].

These patents collectively highlight the continuous advancements in NP technologies for biomedical applications. By refining synthesis methods, enhancing stability through functionalization, and exploring novel applications, these innovations are significantly advancing the clinical utility of NPs. Such progress addresses the critical issue of antibiotic resistance, paving the way for more effective and reliable antimicrobial treatments.

### 10.2. Clinical Trials Involving Nanoparticles

#### 10.2.1. Clinical Trials

Clinical trials are critical for translating the potential of Ag and Au NPs from the laboratory to clinical practice. Recent advancements in clinical trials have highlighted the significant potential of NPs in addressing antibacterial resistance. One notable study explored the clinical efficacy of Argumistin™, a nanosilver-based antibacterial drug. This formulation, containing Ag NPs stabilized with benzyldimethyl[3-(miristoylamino)-propyl]ammonium chloride, was tested in treating infectious diseases in dogs. The trials demonstrated its effectiveness in reducing treatment periods for conditions such as conjunctivitis, gingivitis, and enteritis, making it a promising candidate for human medicine [111]. A separate clinical trial investigated the use of Ag NPs in a controlled setting to assess their impact on wound healing and bacterial infections. The results underscored the NPs ability to enhance antimicrobial activity and promote tissue regeneration, showcasing their dual functionality in therapeutic applications. The study emphasized the importance of NPs size and surface modifications in optimizing their antibacterial properties. [112].

Another critical study focused on the integration of Au NPs in clinical practices for treating drug-resistant bacterial infections. The trial revealed that Au NPs, due to their unique physicochemical properties, could effectively target and disrupt bacterial cell membranes, leading to significant reductions in bacterial viability. This study reinforced the potential of NPs as potent antibacterial agents capable of overcoming the limitations of traditional antibiotics [113].

#### 10.2.2. Drug Delivery

The use of NPs, notably Ag and Au, has gained significant attention in the field of drug delivery for treating bacterial infections. These NPs offer unique advantages due to their small size, large surface area, and the ability to be functionalized with various therapeutic agents. Ag NPs are well-known for their broad-spectrum antimicrobial properties and can be used to deliver antibiotics directly to the site of infection, enhancing the local concentration of the drug and reducing systemic side effects. The functionalization of Ag NPs can be modified to the target compromise toxicity. Clinical trials have shown promising results in using Ag NPs for treating infections caused by resistant bacterial strains [114]. Au NPs have been studied for their potential to enhance the efficacy of antibiotics against multidrug-resistant (MDR) bacteria. A study by Pradeepa et al. demonstrated the successful synthesis of Au NPs using bacterial exopolysaccharides, which were then functionalized with antibiotics such as levofloxacin, cefotaxime, ceftriaxone, and ciprofloxacin. The functionalized Au NPs showed significantly enhanced bactericidal activity against MDR strains of *Escherichia coli*, *Klebsiella pneumoniae*, and *Staphylococcus aureus* compared to free antibiotics. The study highlighted the potential of Au NPs to reduce MIC and MBC of antibiotics, thereby improving their therapeutic efficacy [115]. Comparative studies have shown that both Ag and Au NPs can enhance the delivery and efficacy of antibiotics.

However, each type of NP offers distinct advantages. Ag NPs provide antimicrobial solid activity and can be used effectively in lower concentrations. On the other hand, Au NPs are known for their stability and ease of functionalization, making them suitable for a wide range of therapeutic applications. Despite the promising results, there are concerns regarding the potential toxicity and long-term effects of using these NPs in clinical settings. Studies emphasize the need for a thorough evaluation of the pharmacokinetics, biodistribution, and safety profiles of NP-based drug delivery systems. Ongoing research and clinical trials continue to address these concerns to ensure the safe and effective use of Ag and Au NPs in drug delivery [116].

#### 10.2.3. Multidrug-Resistant

The emergence and proliferation of MDR bacteria pose a significant challenge to modern medicine, necessitating the development of novel therapeutic strategies. Ag and Au NPs, have shown promising antimicrobial properties against MDR pathogens, providing new avenues for treatment. Ag NPs are well-known for their broad-spectrum antimicrobial properties. Desouky et al. evaluated the antibacterial activity of Ag NPs against MDR Gram(−) *bacilli* isolated from clinical samples. The study found that Ag NPs alone, as well as in combination with antibiotics like amikacin and ceftazidime, exhibited potent bactericidal effects. The synergistic action of Ag NPs and antibiotics significantly lowered the MIC and enhanced the antibacterial efficacy, highlighting the potential of Ag NPs in treating MDR bacterial infections [117]. The bactericidal action of Ag NPs involves multiple mechanisms, including oxidative stress induction, protein dysfunction, membrane disruption, and DNA damage. More et al. discussed the detailed mechanisms by which Ag NPs exert their antimicrobial effects, emphasizing their ability to disrupt biofilms and prevent bacterial adhesion. This study highlighted ongoing clinical trials and real-life applications of Ag NPs in medical devices, wound dressings, and coatings for catheters to prevent MDR infections [118].

Au NPs have demonstrated significant potential in enhancing the efficacy of antibiotics against MDR bacteria. A study by Wei et al. developed dual-functional Au NPs functionalized with vascular endothelial growth factor-A165 (VEGF-A165) and (11-mercaptoundecyl)-*N*,*N*,*N*-trimethylammonium (11-MTA). These NPs exhibited both antimicrobial and proangiogenic activities, proving highly effective in treating wounds infected with methicillin-resistant *Staphylococcus aureus* (MRSA) in diabetic mice. The functionalized Au NPs significantly reduced bacterial load and promoted wound healing by enhancing collagen formation and epithelialization [119].

#### 10.2.4. Wound Healing and Infections

The healing of wounds, particularly those complicated by infections, remains a significant challenge in medical practice. Mihai et al. reviewed the use of Ag NPs in wound healing, noting their effectiveness against MDR bacteria and biofilms. Ag NPs promote faster wound closure by modulating inflammatory responses and stimulating reepithelialization. Several clinical trials have demonstrated the efficacy of Ag NP-containing dressings, such as Acticoat, in treating burns and chronic wounds. These trials showed reduced infection rates, enhanced reepithelialization, and overall improved healing outcomes [120]. Boroumand et al. conducted a review of clinical trials involving Ag NPs for wound healing. The review emphasized the broad-spectrum antibacterial properties of Ag NPs, which effectively target antibiotic-resistant bacteria. Clinical trials comparing Ag NP dressings with conventional treatments (e.g., silver nitrate and sulfadiazine creams) revealed that Ag NP dressings significantly reduce bacterial colonization and accelerate wound healing. However, this research also highlighted concerns regarding potential toxicity and the need for further studies to evaluate the long-term safety of Ag NP applications in clinical settings [121].

Au NPs have been investigated for their potential to accelerate wound healing and prevent bacterial infections. Arafa et al. developed thermoresponsive gels containing Au NPs, which demonstrated significant antibacterial activity against *Staphylococcus aureus*, a common pathogen in burn wound infections. These gels exhibited sustained release of Au NPs, promoting effective antibacterial action and enhanced wound healing in burn-infected wounds in animal models. The study highlighted the dual functionality of Au NPs in providing antimicrobial effects and facilitating tissue regeneration [122].

## 11. Regulatory Status of Nanoparticles

The use of Ag and Au NPs in antibacterial applications presents promising opportunities; however, their regulatory status and toxicity remain critical concerns. To ensure the safe and effective use of these NPs, it is crucial to have robust regulatory oversight and conduct comprehensive toxicity assessments.

### 11.1. Toxicity of Nanoparticles

The toxicity of Ag and Au NPs is a significant area of study, as their interaction with biological systems can lead to broad adverse effects. The toxicity of NPs is influenced by their size, shape, concentration, surface charge, and coating materials. For instance, smaller NPs have a larger surface area-to-volume ratio, which can enhance their reactivity and potential toxicity. Similarly, surface modifications can either mitigate or exacerbate their toxic effects. Studies have shown that the physicochemical properties of NPs, such as dissolution rate and surface chemistry, play a crucial role in determining their biological interactions and toxicity profiles [123]. Furthermore, studies have highlighted the high cytotoxicity of silver nanoparticles, particularly in contexts where direct contact with cells occurs. Cazzola et al. demonstrated that silver nanoparticles embedded in titanium surfaces can exhibit marked cytotoxic effects on human osteoblast progenitor cells, with toxicity being influenced by factors such as nanoparticle size and the method of incorporation into the surface. This study underscores the importance of balancing antibacterial efficacy with cytocompatibility to ensure safe biomedical applications [124].

#### 11.1.1. In Vitro and In Vivo Studies

Ferdous et al. showed that Ag NPs exhibit cytotoxic effects on different cell lines. These effects include oxidative stress, DNA damage, and apoptosis, and demonstrated that Ag NPs can induce cytotoxicity in human lung fibroblasts and liver cells [125]. Animal studies have provided insights into the biodistribution and toxicity of NPs. Ag NPs have been shown to accumulate in organs such as the liver, spleen, and lungs, leading to organ-specific toxicity. Chronic exposure to Ag NPs can result in argyria, a condition characterized by skin discoloration due to silver deposition. Studies have shown that Au NPs can induce mild cytotoxicity and oxidative stress in specific cell types, though these effects are often less pronounced than those of Ag NPs [125]. Comparative studies have indicated that Au NPs generally exhibit lower toxicity than Ag NPs; however, while both Ag and Au NPs can penetrate biological barriers and reach various organs, Ag NPs tend to exhibit more pronounced toxicological effects, including oxidative stress and inflammatory responses [123].

#### 11.1.2. Comparative Toxicity

Miyayama et al. (2016) emphasize the importance of understanding the unique interactions between NPs and biological systems. Their research indicates that while both Ag and Au NPs can penetrate biological barriers and reach many human organs, Ag NPs tend to exhibit more pronounced toxicological effects, including oxidative stress and inflammatory responses [123].

### 11.2. Regulatory Framework

Regulatory agencies worldwide have established guidelines to ensure the safety and efficacy of NPs in medical and consumer products. The regulatory status of NPs varies across regions, but common themes include the need for thorough preclinical and clinical evaluations.

#### 11.2.1. United States (FDA)

The U.S. Food and Drug Administration (FDA) regulates NPs under existing frameworks for drugs, biologics, and medical devices. The FDA requires comprehensive toxicological data and clinical trials to evaluate the safety and effectiveness of NPs. For instance, the FDA has approved silver-based wound dressings and coatings for medical devices [126].

#### 11.2.2. European Union (EMA)

The European Medicines Agency (EMA) evaluates NPs under the guidelines for nanomedicines. The EMA emphasizes the need for detailed characterization of NPs, including their physicochemical properties, pharmacokinetics, and toxicological profiles. Products containing NPs must comply with the European Union’s REACH (Registration, Evaluation, Authorization, and Restriction of Chemicals) regulation [127]. The REACH regulation, as amended, specifically addresses the unique properties of nanomaterials, requiring detailed safety assessments and thorough documentation for each nanoform of a substance. The amendments to REACH clarify the registration duties and obligations for nanomaterials, ensuring that all nanoforms are adequately characterized and assessed. This includes providing specific information on particle size, shape, surface area, surface treatment, and other morphological characteristics. Additionally, registrants must provide data on the dissolution rate of nanomaterials in water and relevant biological and environmental media, as well as consider potential toxicological and ecotoxicological impacts [128].

#### 11.2.3. Health Canada

Health Canada employs a working definition for nanomaterials to identify and regulate these substances within existing legislative frameworks. This includes ensuring that nanomaterials used in products such as drugs, medical devices, and consumer products are safe for use and do not pose undue health risks [129].

#### 11.2.4. International Standards

Organizations such as the International Organization for Standardization (ISO) and the Organization for Economic Co-operation and Development (OECD) have developed standards and guidelines for the safe use of NPs. These include protocols for toxicity testing, risk assessment, and environmental impact evaluations [130].

### 11.3. Forward-Looking View

The regulatory landscape for nanoparticles (NPs) continues to evolve as new research sheds light on their complex interactions with biological systems. Key challenges include the development of standardized testing methods, the assessment of long-term exposure risks, and the establishment of clear regulatory pathways for novel NP formulations.

#### 11.3.1. Standardization

Harmonizing regulatory standards across regions is crucial to facilitate the global development and commercialization of NP-based products. Collaborative efforts among regulatory agencies, industry stakeholders, and academic researchers are essential to achieve this goal [130].

#### 11.3.2. Long-Term Studies

Longitudinal studies are needed to understand the chronic effects of NPs on human health and the environment. These studies should focus on the potential for bioaccumulation, persistence, and the impact of NPs on ecosystems [131]. Ongoing research by Health Canada underscores the importance of flexible regulatory approaches that can adapt to emerging scientific knowledge and ensure comprehensive risk assessments [129].

#### 11.3.3. Regulatory Pathways

Developing clear regulatory pathways for emerging NPs technologies will help streamline the approval process while ensuring safety and efficacy. This includes fostering innovation through regulatory science initiatives and adaptive regulatory frameworks [128]. The FDA and EMA are working towards integrating advanced scientific tools and methods into their regulatory processes to assess the risks and benefits of NPs better [126,127].

By addressing these challenges and advancing our understanding of NP toxicity, the scientific community can ensure the safe and effective use of Ag and Au NPs in antibacterial applications, ultimately contributing to the global effort to combat antibiotic-resistant bacteria.

## 12. Conclusions

The findings of our research are significant as they reveal that spherical Ag NPs demonstrate a higher antibacterial activity against Gram(−) bacteria than against Gram(+) bacteria. The antibacterial efficacy of Ag NPs is more influenced by their size than by their shape, with particles smaller than 20 nm being particularly effective. In contrast, the shape of Au NPs significantly influences their antibacterial activity, with non-spherical shapes such as peanut, rod, porous spherical, and nanoclusters being particularly effective. The synthesis of non-spherical Au NPs improves their antibacterial capacity, achieving increased effectiveness against *E. coli* and *S. aureus*, with minimum inhibitory concentrations (MIC) and minimum bactericidal concentrations (MBC) comparable to or even higher than those obtained with spherical Ag NPs. However, the synthesis of non-spherical Au NPs involves a more complex process. Generally, Ag NPs show antimicrobial activity against a broader spectrum of bacterial strains Gram(+/−) compared to Au NPs.

Moreover, the relentless progress in nanoparticle technologies, as evidenced by a multitude of patents, underscores the pioneering work in enhancing synthesis methods, stability, and antibacterial efficacy of Ag and Au NPs. These patents, which cover synthesis methods, functionalization techniques, and applications in drug delivery systems and medical devices, are pivotal in the fight against antibiotic resistance and in improving clinical outcomes. The evolving regulatory landscape further underscores the need for robust oversight and comprehensive toxicity assessments. Addressing these regulatory challenges, which include detailed characterization, preclinical and clinical evaluations, and long-term exposure assessments, is crucial. By doing so and deepening our understanding of NP toxicity, the scientific community can ensure the safe application of Ag and Au NPs in antibacterial treatments, thereby contributing significantly to the global effort to combat antibiotic-resistant bacteria.

This review finds wide-ranging applications of Ag and Au NPs with antibacterial effects. However, determining an optimal concentration for a specific effect is challenging due to the lack of standardization in the reported protocols and the extensive variety of nanomaterial characteristics. Standardized experimental protocols are crucial to obtaining consistent and comparable results, which will enhance the reproducibility and reliability of research findings in this field.

Finally, it is of utmost importance to emphasize that the antibacterial effectiveness of both Au NPs and Ag NPs varies among different bacterial strains, and this variability is also associated with the synthesis method employed. Therefore, standardizing synthesis methods and quantitative reporting of MIC and MBC values is crucial. Such standardization will not only enhance the comparability and reproducibility of research findings but also facilitate the integration of advanced data analysis techniques, including artificial intelligence, which relies on consistent and quantitative data. This approach is essential for optimizing NPs properties for specific antibacterial applications and accelerating the development of effective NP-based antibacterial treatments. Moreover, the availability of high-quality, standardized data is critical for developing robust AI models. Researchers should share their data in accessible repositories, ensuring it is available for AI training and validation. By collaborating to build comprehensive datasets, we can significantly enhance the predictive power of AI models, paving the way for groundbreaking innovations in NP-based antibacterial treatments.

## Figures and Tables

**Figure 1 pharmaceuticals-17-01134-f001:**
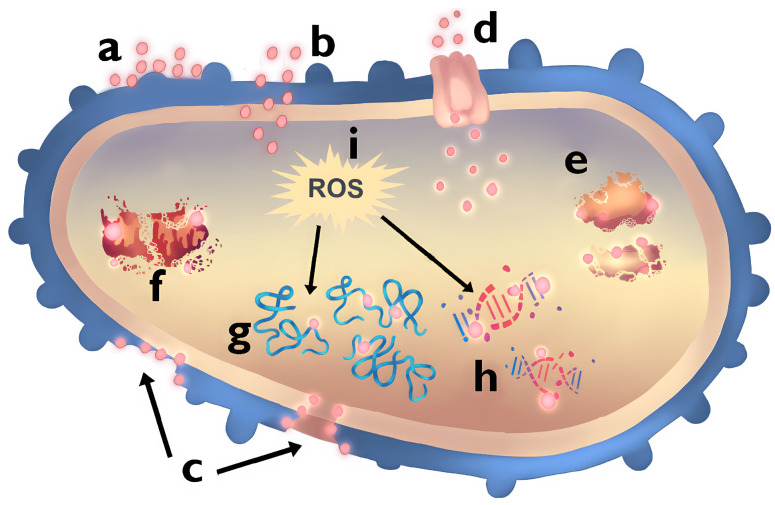
Mechanisms of antibacterial action of Ag NPs include: (**a**) adhesion of Ag NPs to the cell wall, (**b**) cross the membrane; (**c**) damage to the cell wall and membrane induced by Ag NPs; (**d**) entry of Ag NPs into the bacterial cell is facilitated by porin proteins, which serve as conduits for their passage; (**e**) disassembly of ribosomes triggered by Ag NPs; (**f**) mitochondrial dysfunction caused by Ag NPs; (**g**) protein denaturation induced by Ag NPs; (**h**) DNA damage inflicted by Ag NPs; and (**i**) reactive oxygen species (ROS) production and oxidative stress induction resulting from Ag NPs interaction. Modified from [25].

**Figure 2 pharmaceuticals-17-01134-f002:**
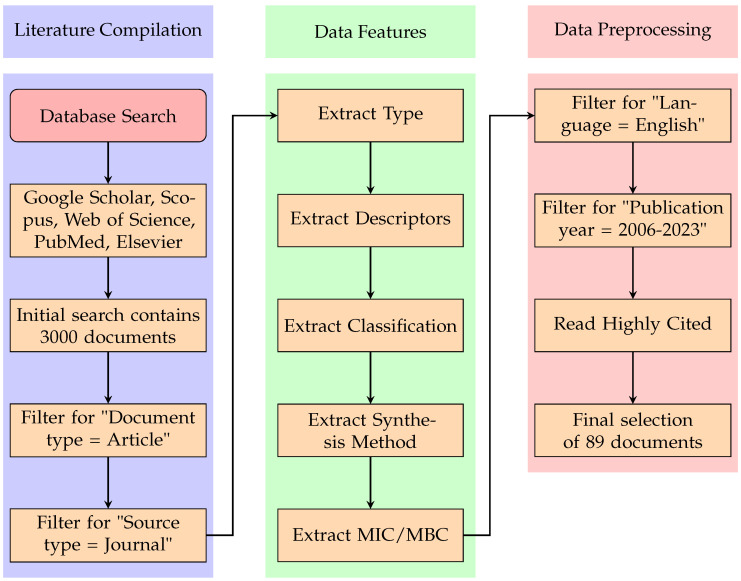
Flowchart detailing the process of selection and analysis of articles for the review. The methodology focused on identifying sources reporting the use of NPs against Gram(+/−) bacteria in concentration quantitative form. Specific inclusion criteria addressed those studies that detailed the synthesis, size, and shape, as well as the minimum inhibitory concentration (MIC) and minimum bactericidal concentration (MBC) of the NPs.

**Figure 3 pharmaceuticals-17-01134-f003:**
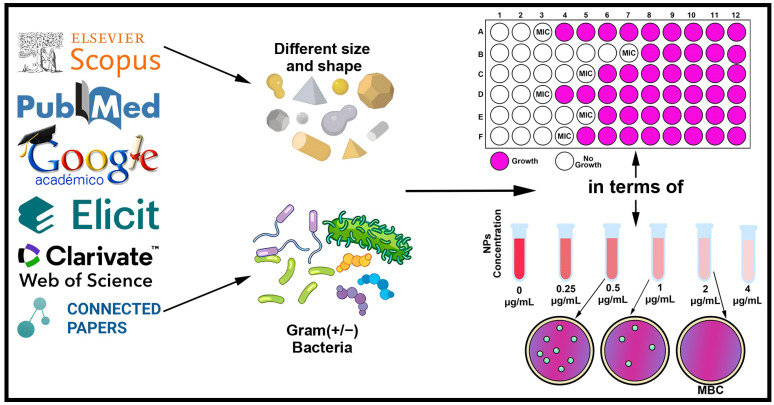
Outline of the article selection and analysis process for the review. The methodology focused on identifying sources reporting the use of NPs against Gram(+/−) bacteria. Specific inclusion criteria addressed those studies that detailed the synthesis, size, and shape, as well as the minimum inhibitory concentration (MIC) and minimum bactericidal concentration (MBC) of the NPs.

**Figure 4 pharmaceuticals-17-01134-f004:**
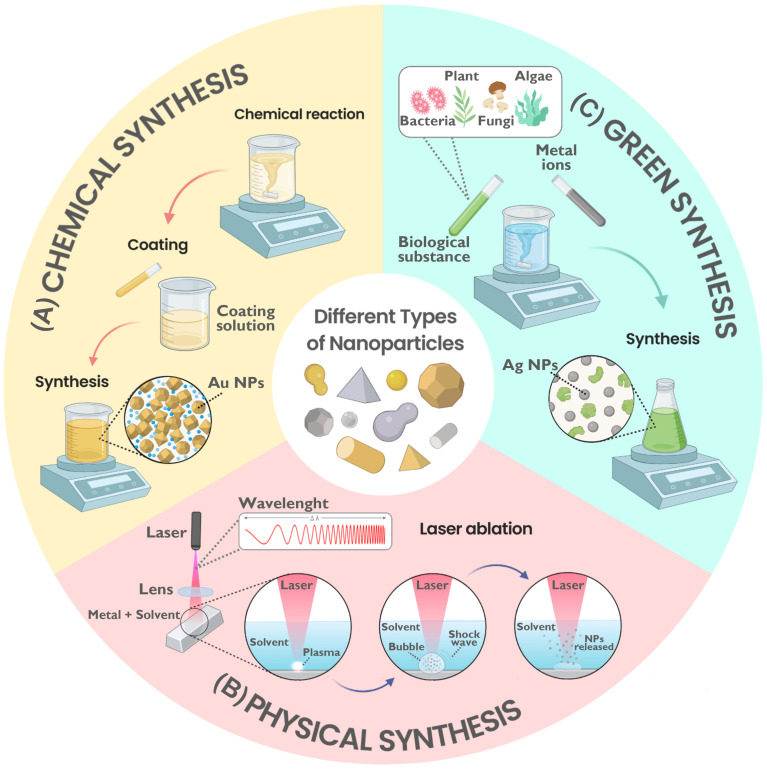
Synthesis methods of NPs. This figure depicts the three primary methods for synthesizing NPs: (A) Chemical synthesis involves the reduction of metal salts using chemical reducing agents; (B) Physical synthesis includes methods such as laser ablation, which involves breaking down bulk metals into NPs and (C) Green synthesis utilizes biological organisms or plant extracts to reduce metal ions into NPs. Each method influences the size, shape, and surface properties of the NPs, which in turn affect their antibacterial activity.

**Figure 5 pharmaceuticals-17-01134-f005:**
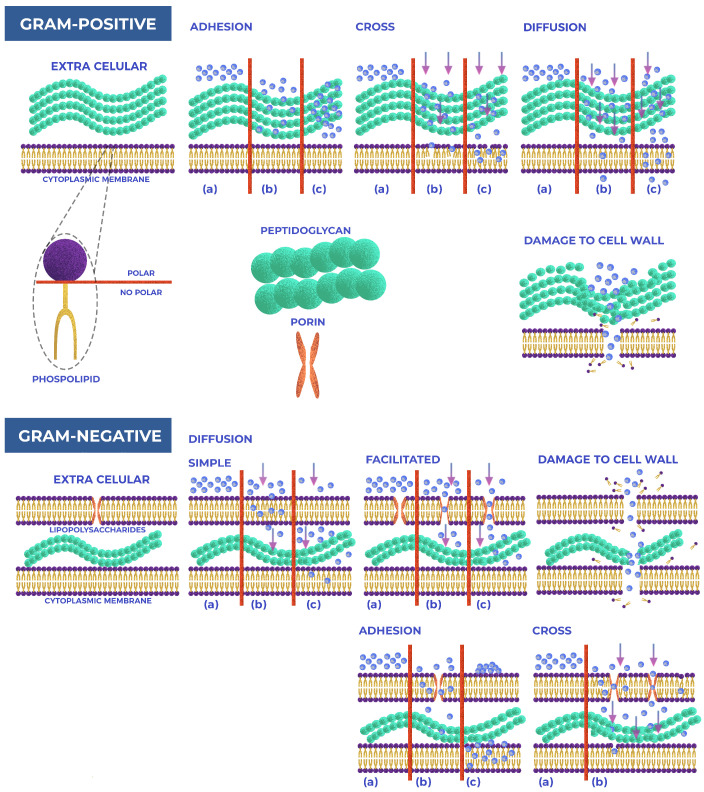
Gram(+) and Gram(−) bacterial cell membranes with the mechanisms of action of nanoparticles. The arrows indicate how the NPs pass through the different layers of the cell wall. **Diffusion**: NPs pass through the cell wall layer according to the concentration gradient: *Diffusion in Gram(+)*: (a) NPs are highly concentrated in the extracellular space. (b) NPs begin to infiltrate the cell wall, with increasing concentration inside. (c) NPs are distributed throughout the cell wall and cytoplasmic membrane, showing higher intracellular concentration. *Simple Diffusion in Gram(−)*: (a) NPs are highly concentrated in the extracellular space. (b) NPs begin to infiltrate the outer membrane, showing a gradient of decreasing concentration from outside to inside. (c) NPs are distributed throughout the periplasmic space, indicating a balanced concentration between the extracellular and intracellular environments. *Facilitated Diffusion in Gram(−)*: (a) NPs attach to the outer membrane at the porins. (b) NPs begin to pass through the porins into the periplasmic space. (c) NPs move through the porins and spread throughout the periplasmic space. **Adhesion**: NPs adhere to bacterial cell wall. For *Gram(+)* bacteria, NPs interact with peptidoglycans, attaching at multiple sites (a, b, c) to enhance antibacterial effects. In *Gram(−)* bacteria, NPs attach to lipopolysaccharide (LPS) layer and outer membrane at various adhesion points (a, b, c). **Cross**: NPs cross the bacterial cell wall structures: *Gram(+)* bacteria: (a) NPs initially interact with and infiltrate the peptidoglycan layer. (b) NPs traverse the peptidoglycan, moving deeper towards the cytoplasmic membrane. (c) NPs continue to move through the cell wall structures, approaching the cytoplasmic membrane. *Gram(−)* bacteria: (a) NPs initially interact with and infiltrate the LPS layer. (b) NPs traverse the outer membrane, moving towards the periplasmic space. (c) NPs continue to move through the cell wall structures, approaching the cytoplasmic membrane. **Damage to Cell Wall**: In both *Gram(+/−)* bacteria, the interaction of NPs results in significant damage to the cell wall. This disruption of membrane integrity and cellular functions leads to bacterial cell death.

**Figure 6 pharmaceuticals-17-01134-f006:**
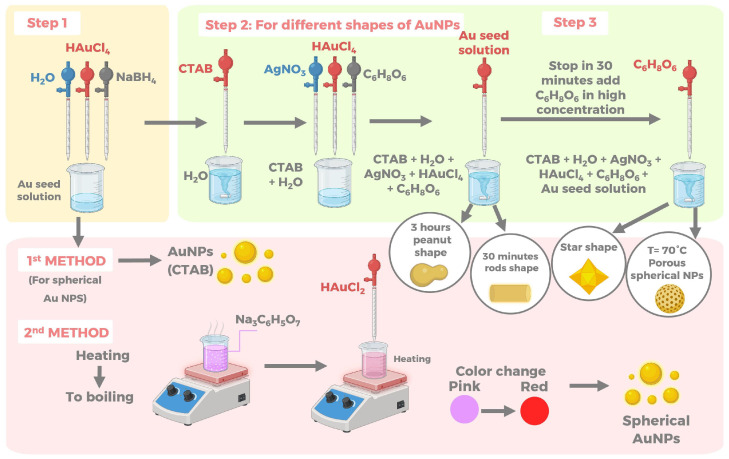
Synthesis methodology for various shapes of Au NPs as described by Piktel et al. [56]. The process consists of two main steps: First, preparation of Au seed solution using CTAB, water, HAuCl_4_, and NaBH_4_. Second, obtaining different shapes (rod, star, peanut, and porous spherical) by mixing CTAB, water, AgNO_3_, HAuCl_4_, C_6_H_8_O_6_, and the Au seed solution with varying conditions. Additionally, two methods for synthesizing spherical Au NPs are depicted: one using CTAB and the other involving heating a trisodium citrate solution.

**Table 1 pharmaceuticals-17-01134-t001:** WHO (2024) list of priority antibiotic-resistant pathogens [4].

Priority	Bacterium	Resistant Antibiotics
Critical	*Acinetobacter baumannii*	Carbapenems
Critical	*Pseudomonas aeruginosa*	Carbapenems
Critical	*Enterobacteriaceae*	Carbapenems, Cephalosporins
Critical	*Mycobacterium tuberculosis*	Rifampicin
High	*Enterococcus faecium*	Vancomycin
High	*Staphylococcus aureus*	Methicillin, Vancomycin
High	*Helicobacter pylori*	Clarithromycin
High	*Campylobacter* spp.	Fluoroquinolones
High	*Salmonellae*	Fluoroquinolones
High	*Neisseria gonorrhoeae*	Cephalosporins, Fluoroquinolones
High	*Pseudomonas aeruginosa*	Carbapenems
Medium	*Streptococcus pneumoniae*	Penicillin, Macrolides
Medium	*Haemophilus influenzae*	Ampicillin
Medium	*Shigella* spp.	Fluoroquinolones

**Table 2 pharmaceuticals-17-01134-t002:** Comparison of synthesis methods for Ag and Au NPs and their impact on antibacterial properties.

Synthesis Method	Advantages	Disadvantages	Impact on NPs Features and Antibacterial Activity	References
Physical	High purity NPs. Large quantities.	High energy requirement. Special equipment. Specific conditions needed.	Homogeneous size and shape. High interaction with bacteria. Suitable for high-tech applications.	[41,45]
Chemical	Scalable. Cost-effective.	Potentially toxic reagents. Residual solvents.	Controllable size and shape. Variable surface modifications. Effective in various antibacterial assays.	[47,48]
Green	Eco-friendly. Low toxicity. Cost-effective.	Complexity due to biological components. Variability NP properties.	High solubility and stability. High effectiveness against a broad range of Gram(+/−) bacteria.	[13,45,49]

**Table 3 pharmaceuticals-17-01134-t003:** Summary of data collection and antibacterial activity ranges for Ag and Au NPs.

Nanoparticle	Data Collected	MIC (μg/mL)	MBC (μg/mL)
Ag	Total: 458; MIC: 318, MBC: 140	0.11–1200	0.22–1500
Au	Total: 300; MIC: 200, MBC: 100	0.00008–8000	0.00008–16,000

**Table 4 pharmaceuticals-17-01134-t004:** Antibacterial effectiveness of Ag and Au NPs with varying sizes and bacterial strains.

Nanoparticle	Size (nm)	Shape	Synthesis Method	Bacterial Strain	MIC (μg/mL)	MBC (μg/mL)	Reference
Ag	5	Spherical	Chemical	*A. actinomycetemcomitans*	6–50	-	[51]
	5	Spherical	Chemical	*E. coli* MTCC 443	20–110	-	[10]
	7			*E. coli* MTCC 739	60–160	-	
	10			*B. subtilis* MTCC 441	30–120	-	
	15			*S. aureus* NCIM 5021	70–200	-	
	15	Spherical	Chemical	*F. nucleatum*	12–50	-	[51]
	55	Spherical	Chemical	*S. sanguis*	100–200	-	
				*S. mutans*		-	
				*S. mitis*		-	
				*E. coli*		-	
Au	25	Spherical/Stars	Chemical	*S. mutans*	0.97–3.17	1.95–6.46	[55]
				*S. sanguinis*			
				*S. salivarius*			
	60	Spherical/Stars	Chemical	*S. mutans*	91.61–148.21	125–289.28	
				*S. sanguinis*			
				*S. salivarius*			
	90	Spherical/Stars	Chemical	*S. mutans*	232.95–500	217.26–1000	
				*S. sanguinis*			
				*S. salivarius*			

## Data Availability

The data presented in this study are available on request.

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
