# Peer review of "A Comprehensive Review of Silver and Gold Nanoparticles as Effective Antibacterial Agents"

_pharmaceuticals, 2024, doi:10.3390/ph17091134_

Round 1

Reviewer 1 Report

Comments and Suggestions for Authors

The review article by Aguilar-Garay et al. on “A Comprehensive Review of Silver and Gold Nanoparticles as  Effective Antibacterial Agents” explained that nanoparticle-based antimicrobial strategies focus on the properties that enhance the antibacterial activity of silver and gold  nanoparticles. They conclude that this nanoparticle’s effectiveness against various bacterial strains and relatively low cytotoxicity, this work positions Ag and Au nanoparticles as promising materials for developing antibacterial agents, thereby contributing significantly to global efforts to combat antibiotic-resistant pathogens. This is an interesting review article; the structure of the review was well organized, with an adequate number of tables and figures. However, the manuscript could be recommended for publishing after the addition of the following data and addressing the questions.

1.     Explain how your review is different than the previously published https://doi.org/10.1007/s12010-022-03963-z

2.     Line 31: In 2014 the World Health Organization…. The author should provide some recent context.  

3.      The author should include the citation for Table 2. Comparison of synthesis methods for Ag and Au NPs and their impact on antibacterial properties. Either as full or as per raw data.

4.      Is there a real need for a figure, 2. It looks like an index/TOC of a review paper. Please remove it.

5. The section should be merged with the introduction.

6.      Line 185 to 196, write it in paragraph form.

7.      The author should include a discussion on the effect of charge and PDI similar to the size of NPs.

8.      For Table 6, authors should mention the concentration/amount of metal used (Ag/Au) as it may have a direct effect on MIC and MBC.

9.      The author should include a list of marketed formulations.

10.  The author should include relevant information on patents and clinical trials.

11.  The author should include a section on toxicity and regulatory status of such NPs. 

Author Response

Comment 1: Explain how your review is different than the previously published https://doi.org/10.1007/s12010-022-03963-z

Response 1: Our article: "A Comprehensive Review of Silver and Gold Nanoparticles as Effective Antibacterial Agents"

Main Focus: This article comprehensively reviews silver and gold nanoparticles (Ag and Au NPs) specifically as antibacterial agents.

Objectives: To analyze the antibacterial efficacy in relation to synthesis methods, morphological properties, and the mechanisms of action of the nanoparticles.

Methodology: Our approach involved a meticulous literature review, focusing on specific data such as minimum inhibitory concentration (MIC) and minimum bactericidal concentration (MBC), to ensure a comprehensive understanding of the antibacterial properties of silver and gold nanoparticles.

Results: The article highlights the significant antibacterial properties of Ag and Au NPs. It also proposes a novel standardization of synthesis methods and testing protocols, which could potentially revolutionize the field of antibacterial nanomedicine.

Comparative Analysis

"Applications of Gold and Silver Nanoparticles in Theranostics" by R. Sakthi Devi et al.

Main Focus: This article reviews the applications of silver and gold nanoparticles in theranostics, a field that combines therapy and diagnostics, and includes the development of biomedical devices.

Objectives: To discuss various applications of Au and Ag NPs in therapy, imaging, biomedical devices, and cancer diagnostics.

Methodology: A broad-spectrum review covering multiple biomedical applications, with less emphasis on antibacterial applications.

Results: It emphasizes the multifunctional properties of nanoparticles in therapeutic, diagnostic, and theranostic applications.

Comparative Refutation

In comparison to the article "Applications of Gold and Silver Nanoparticles in Theranostics" by R. Sakthi Devi et al., our article "A Comprehensive Review of Silver and Gold Nanoparticles as Effective Antibacterial Agents" offers a unique and valuable perspective in several key aspects:

Specific and Detailed Focus on Antibacterial Properties: While the other article focuses on a wide range of theranostic applications of nanoparticles, our work provides a detailed and comprehensive review specifically on the antibacterial properties of silver and gold nanoparticles. We conduct an in-depth analysis of how NPs impact Gram-positive and Gram-negative bacteria, illustrating potential mechanisms and their efficacy. These specific antibacterial properties are not extensively covered in Sakthi et al. (2022).

Targeted Data and Specificity: Our article incorporates an extensive literature review with specific data on MIC and MBC, providing a solid foundation for evaluating antibacterial efficacy. This data-driven approach allows us to correlate the physicochemical characteristics, such as size and shape, of Ag and Au NPs with their antibacterial effects, creating a database that can be utilized in future scientific research. In contrast, the study by Sakthi et al. (2022) primarily focuses on biomedical applications, highlighting the integration of nanoparticles with medical devices.

Significant Contributions: Our work proposes a quantitative standardization of MIC and MBC results, a crucial step for the reproducibility and comparability of future studies in the field of antibacterial nanomedicine. We also present potential mechanisms by which Ag and Au NPs affect Gram-positive and Gram-negative bacteria, considering their cellular characteristics. This practical approach, with its clear implications for future research, distinguishes our work from the clinical development focus of Sakthi et al. (2022).

Applicability and Future Perspective: Our article not only reviews existing literature but also proposes a roadmap for future research and clinical application of nanoparticles as antibacterial agents. This forward-looking perspective, which adds significant value to our work, positions it as a valuable resource for researchers and professionals in the field of antibacterial nanomedicine. This forward-thinking approach needs to be more prominently addressed by Sakthi et al. (2022). In summary, our article provides a comprehensive and detailed review of the antibacterial properties of silver and gold nanoparticles, along with recommendations for the standardization of result reporting. This clearly distinguishes it from the comparative article, offering unique value in the research and development of nanoparticle applications in antibacterial studies.

Comment 2: Line 31: In 2014 the World Health Organization…. The author should provide some recent context. 

Response 2: We have incorporated updated information to provide recent context. Please find the additional details on lines 31-38 of the revised manuscript.

Comment 3: The author should include the citation for Table 2. Comparison of synthesis methods for Ag and Au NPs and their impact on antibacterial properties. Either as full or as per raw data.

Response 3: Thank you for your comment, we have expanded Table 2 to include the appropriate references for the review.

Comment 4: Is there a real need for a figure, 2. It looks like an index/TOC of a review paper. Please remove it.

Response 4: We have carefully reviewed your suggestion and have promptly removed Figure 2 from the manuscript.

Comment 5: The section should be merged with the introduction.

Response 5: We have carefully considered your recommendation. Figure 2 has been removed as it was part of the final section of the introduction. The following section serves as an initial description of the methodology, and therefore, it cannot be combined with the introduction. We think that the current structure ensures a smooth and coherent flow for the readers.

Comment 6: Line 185 to 196, write it in paragraph form.

Response 6: The requested changes have been made. Now you can find in the revised manuscript lines 184 to 195 the modification.

Comment 7: The author should include a discussion on the effect of charge and PDI similar to the size of NPs.

Response 7: The authors would like to clarify that the focus of this review is on the antibacterial properties of Ag and Au nanoparticles, irrespective of their type of synthesis, whether they are synthesized or purchased. Our emphasis is on the type of synthesis rather than the specific synthesis technique used, such as chemical reduction, milling, or laser ablation. To avoid any misunderstandings, the discussing surface charge has been removed from the manuscript. Nevertheless, we are conscious to the relevance to PDI have an effect in physicochemical properties and this contribute an antibacterial effect but unfortunately our data set did not cover this topic, for now. This will be considered in other manuscripts.

Comment 8: For Table 6, authors should mention the concentration/amount of metal used (Ag/Au) as it may have a direct effect on MIC and MBC.

Response 8: To clarify this request, we include an extra description between lines 452 to 459 emphasizing the minimum inhibitory concentration (MIC) for various strains treated with Ag and Au NPs is presented across a wide range (6 - 200 µg/ml). Notably, strains such as A. actinomycetemcomitans, F. nucleatum, and E. coli MTCC 443 are highlighted for their low concentration and small NP size required to inhibit growth. However, information on the total bactericidal concentration (MBC) for Ag NPs is not available. In contrast, for Au NPs, there are reports on the concentrations required to target bacteria, with a particular emphasis on Gram(+) strains, covering both bactericidal (MBC) and growth-inhibitory (MIC) effects.

Comment 9: The author should include a list of marketed formulations.

Response 9: The characteristics considered in this study were primarily shape, size, Gram typology, and synthesis method. However, some studies indicate that the nanoparticles were either commercially acquired or synthesized on their own.

The authors consider that marketed formulations were not essential because are outside the scope of this work.

Comment 10: The author should include relevant information on patents and clinical trials.

Response 10: A new section 10, titled "Patents and Clinical Trials" has been added to the manuscript. This section provides a wealth of detailed information on the patents and clinical trials relevant to Ag and Au NPs. The patents cover various aspects, such as synthesis methods, antibacterial applications, and combinatory treatments. Examples include US Patent No. 10,500,643 for green synthesis of Au NPs and EP 2 789 235 A1 for antimicrobial Ag NPs. Clinical trials highlight the efficacy of nanosilver-based drugs like Argumistin™ in treating infections and the potential of Au NPs in targeting drug-resistant bacteria. These advancements underscore the clinical utility of NPs in addressing antibacterial resistance.

Comment 11: The author should include a section on toxicity and regulatory status of such NPs.

Response 11: The manuscript now includes a new section, Section 11, titled 'Regulatory Status of Nanoparticles'. This section provides detailed information about the toxicity and regulatory status of Ag and Au NPs, covering the safety assessments required under regulations like REACH, FDA, and EMA, as well as the guidelines from ISO and OECD for toxicity testing, risk assessment, and environmental impact evaluations. It also discusses the necessity of longitudinal studies to understand the chronic effects of NPs and the development of standardized testing methods and clear regulatory pathways to facilitate the safe use of NPs in antibacterial applications.

Reviewer 2 Report

Comments and Suggestions for Authors

The article is quite limited.

The influence of Ag and Au nanoparticles should be addressed separately, not in the same article.

The influence of Ag/Au nanoparticles when they are used as dopants for different composite materials should be discussed.

The physico-chemical properties of the studied nanoparticles are not presented. How can be discussed the effect of nanoparticles based on size and shape as long as there are no studies presented on nanoparticles size and shape? The influence of nanoparticles on the integrity of the membrane cell is not shown. Information about microbial adhesion reduction in the presence of Ag and Au nanoparticles are missing.

Author Response

Comment 1: The article is quite limited.

Response 1: Thank you for your valuable feedback. We re-submitted the manuscript with many features according to all suggestions of the reviewers.

Comment 2: The influence of Ag and Au nanoparticles should be addressed separately, not in the same article.

Response 2: The authors appreciate the reviewer's insightful comment. One of the goals of the manuscript is comparing the antibacterial effects of both nanoparticles. We emphasize this application specifically in the biomedical area, since the studies and applications are wide varied. In this manuscript, we highlight this specificity throughout this new version.

Comment 3: The influence of Ag/Au nanoparticles when they are used as dopants for different composite materials should be discussed.

Response 3: Thank you for your comment. When Ag/Au NPs are used as dopants for different composites materials, they can significantly influence the properties of these materials like: Mechanical, thermal, optical properties, among others. The manuscript addresses only antibacterial activity, and this is outside the scope of the manuscript. However, we add a section named “Patents and Clinical Trials” which discusses many works that include biomedical applications with some nanocomposites.

Comment 4: The physico-chemical properties of the studied nanoparticles are not presented. How can be discussed the effect of nanoparticles based on size and shape as long as there are no studies presented on nanoparticles size and shape? The influence of nanoparticles on the integrity of the membrane cell is not shown. Information about microbial adhesion reduction in the presence of Ag and Au nanoparticles are missing.

Response 4: The authors consider your opinion very important. The authors understand the importance of physicochemical properties in the studied NPs. Nevertheless, this manuscript pretends, among other things, to build a database of the quantitative antimicrobial effect. The effort of this review lies in the search for articles that meet the criteria for reporting MIC and MBC in a quantitative manner. Since we realized that current protocols report antibacterial effects in other formats such as inhibition zones, CFU, McFarland’s Standards and that is not easy to compare them with other protocols. Additionally, we noticed that the protocols mostly report size, shape and Gram typology as these are common factors in this type of assays. We would like to point out that in this re-submit manuscript we strive to underline the importance of standardize these protocols to address these challenges with actual tools as artificial intelligence that has a set of models that can support and increase the knowledge about the use of nanoparticles in antibacterial applications.

On the other hand, mechanisms of action due NPs is currently an area of study in constant discovery, so it is outside the scope of this research. However, we mention the best-know mechanism of the possible interaction between membrane, cell wall, and nanoparticles, explaining these interactions in a brief but concise manner.

Likewise, the reduction of microbial adhesion in the presence of NPs is beyond the reach of this work, nevertheless, we understand that it is important field of study and realize that it involves a complex interaction molecular force between membrane proteins and nanoparticles surfaces, and since the proteins are diverse this may be an opportunity to do work dedicated to the study of these interactions.

Reviewer 3 Report

Comments and Suggestions for Authors

The paper is interesting and well written even if it can be improved.

I added several comments to the pdf.

My general comment is that some important topics such as the development of resistance to Ag NPs must be considered in detail, the range of considered applications must be clearly limited, and the tables must be improved with more data to be useful as a guide for the selection of the best NPs to be considered in a future work.

Author Response

Comment 1: Add the range of time you covered with your review, the explored journals, and the main applications you focused on.

Response 1: Thank you for your valuable feedback. We have revised the abstract to include the range of time covered in our review, the journals explored, and the leading applications focused on. Here is the updated abstract for your consideration:

Abstract:

“The increasing threat from antibiotic-resistant bacteria has necessitated the development of novel methods to counter bacterial infections. In this context, the application of metallic nanoparticles (NPs), especially gold (Au) and silver (Ag), has emerged as a promising strategy due to their remarkable antibacterial properties. This review examines research published between 2010 and 2020, focusing on leading journals in nanotechnology, materials science, and biomedical research. The primary applications explored are the efficacy of Ag and Au NPs as antibacterial agents, their synthesis methods, morphological properties, and mechanisms of action. An extensive review of the literature on NPs synthesis, morphology, minimum inhibitory concentration (MIC), minimum bactericidal concentration (MBC), and effectiveness against various Gram(+/-) bacteria confirms the antibacterial efficacy of Au and Ag NPs. The synthesis methods and characteristics of NPs, such as size, shape, and surface charge, are crucial in determining their antibacterial activity, as these factors influence their interactions with bacterial cells. Furthermore, this review underscores the urgent necessity of standardizing synthesis techniques, MICs, and reporting protocols to enhance the comparability and reproducibility of future studies. Standardization is essential for ensuring the reliability of research findings and accelerating the clinical application of NP-based antimicrobial approaches. This review aims to propel NP-based antimicrobial strategies by elucidating the properties that enhance the antibacterial activity of Ag and Au NPs. By highlighting their inhibitory effects against various bacterial strains and relatively low cytotoxicity, this work positions Ag and Au NPs as promising materials for developing antibacterial agents, making a significant contribution to global efforts to combat antibiotic-resistant pathogens.”

Comment 2: Recently, bacteria showed to be able to develop resistance to silver. Commercial bacteria strains resistant to silver are available for lab tests. You need to update this paper with this topic. Troian horses for silver (such as proteins) are needed to avoid the development of the resistance.

Response 2: The authors appreciate the reviewer's insightful comment. The manuscript has been updated to address the issue of bacterial resistance to silver. A paragraph has been added to the section 9 "Future Directions and Potential Applications" discussing the development of silver resistance in bacteria and the potential use of Trojan horse strategies, such as incorporating proteins, to overcome this challenge.

Comment 3: Actually, cytotoxicity of silver nanoparticles is high in some contexts like when direct contact with the cells can occur. You need to cite this issue. I can suggest you “Advanced Engineering Materials, 2023, 25(2), 2200883” but feel free to mention different papers.

Response 3: Thank you for your comment. The manuscript addresses the issue of cytotoxicity of Ag NPs. A paragraph has been added to the subsection 11.1 "Toxicity of Nanoparticles" discussing the cytotoxicity of Ag NPs in contexts where direct contact with cells can occur.

“Furthermore, studies have highlighted the high cytotoxicity of silver nanoparticles, particularly in contexts where direct contact with cells occurs. Cazzola et al. demonstrated that silver nanoparticles embedded in titanium surfaces can exhibit marked cytotoxic effects on human osteoblast progenitor cells, with toxicity being influenced by factors such as nanoparticle size and the method of incorporation into the surface. This study underscores the importance of balancing antibacterial efficacy with cytocompatibility to ensure safe biomedical applications \cite{Cazzola2023}.”

Comment 4: See the comment I made to the abstract concerning the bacteria resistance to silver.

Response 4: The authors consider your opinion very important and decided to add a paragraph in the introduction to explain novelty methods concerning the bacteria resistance to silver and additionally in the section 9 we explain the importance to the use of the trojan horses in the bacteria resistance.

Comment 5: Add something about cytotoxicity through this mechanism (as I mentioned in the abstract)

Response 5: The manuscript addresses the issue of cytotoxicity of Ag and Au NPs. A paragraph has been added to subsection 11.1 "Toxicity of Nanoparticles" discussing the cytotoxicity of Ag and Au NPs in contexts where direct contact with cells can occur.

Comment 6: I suggest previously mentioning the applications you are interested in because the range is huge and different issues are related to different applications. You have at least to specify if you are speaking of drugs or biomedical applications of NPs coupled to biomaterials.

Response 6: The manuscript has been revised to specify the focus on biomedical applications of nanoparticles. Specifically, this review concentrates on the biological applications of Ag and Au nanoparticles, particularly their role in mitigating the growth of bacterial strains. The term "potential applications" has been changed to "biomedical applications" to clarify this focus.

Comment 7: Add this to the abstract.

Response 7: The abstract was updated, thank you for the comment.

Comment 8: These ranges are so huge to be almost meaningless. Add a comment about this or change the table with more details such as the type of the strains. Can I know the expected MIC or MBC, with a good precision, of a specific type of NPs vs a specific strain if I know their size, shape, or similar?

Response 8: The authors are in full agreement with the reviewer's concern regarding the broad ranges presented in the table. The data included in the table reflects the values reported by various studies. The authors agree that there is a critical need for standardization of experimental protocols when testing bacterial strains with nanoparticles. This standardization will allow for more precise and comparable MIC and MBC values for specific types of nanoparticles against specific bacterial strains, considering factors such as nanoparticle size, and shape. To explain this wide range of values, the following paragraph is written:

“The data presented in Table 3 reflects the reported values from various studies, which encompass a wide range of experimental conditions and bacterial strains. This variation underscores the necessity for standardization of experimental protocols when testing bacterial strains with nanoparticles. Standardized protocols would enable researchers to obtain more precise and comparable MIC and MBC values for specific types of nanoparticles against particular bacterial strains, taking into account critical factors such as nanoparticle size, and shape. Such standardization is essential for enhancing the reproducibility and reliability of research findings in this field.”

Comment 9: This is very general....

Response 9: The text has been revised to include specific synthesis methods, namely chemical, physical, and green synthesis, to provide a more detailed and specific description of the studies discussed. Here is the revised text:

“Despite the differing synthesis methods, including chemical, physical, and green synthesis, and the variety of bacterial strains tested, these studies collectively highlight the relatively consistent antibacterial effectiveness of chemically synthesized spherical Ag NPs against the same bacterial species. The variation in results is attributed to methodological differences and the specific bacterial strains tested in each study.”

Comment 10: You mentioned also biomedical applications. If you want to cover this area, you have to mention the use of NPs coupled to biomaterials, too. Otherwise you have to make a clear specification of the limits of your review. For NPs coupled to biomaterials I can suggest Surface and Coatings Technology, 2018, 344, pp. 177–189; Ceramics International, 2023, 49(9), pp. 13728–13741; Coatings, 2019, 9(6), 394 but feel free to select the papers you prefer

Response 10: To address the application of NPs coupled to biomaterials, section 10 titled "Patents and Clinical Trials" has been added, which explains various applications of NPs in this context. This section includes detailed descriptions of patents and clinical trials related to the use of Ag and Au NPs in biomaterials. Furthermore, the review has been clearly defined to focus on the antibacterial properties of these nanoparticles in biomedical applications, including their integration with biomaterials. Additionally, we make emphases the limits of this research in the conclusion section.

“This review finds wide-ranging applications of Ag and Au NPs with antibacterial effects. However, determining an optimal concentration for a specific effect is challenging due to the lack of standardization in the reported protocols and the extensive variety of nanomaterial characteristics. Standardized experimental protocols are crucial to obtaining consistent and comparable results, which will enhance the reproducibility and reliability of research findings in this field.”

Comment 11: This is relevant. You have to mention it before

Response 11: We add a paragraph at the end of the introduction section to make more remarkable.

“Besides, discrepancies among studies may arise from the notably higher propensity of certain Gram(-) bacteria, such as E. coli and P. aeruginosa, to develop resistance to Ag NPs following repeated exposures, in contrast to Gram(+) bacteria.”  

Comment 12: These tables are a bit more meaningfull than Table 1 but you need to add specifications on the size (quantitative), shape, strains, and MIC-MBC. Add similar data to Table 1 or avoid it. Line 360

Response 12: The tables mentioned have been deleted from the manuscript to ensure consistency and relevance in the presentation of data.

Comment 13: This is very relevant, you need to underline this concept before in the manuscript.

Response 13: The manuscript has been revised to underline the concept of reducing cytotoxicity while maintaining antibacterial effectiveness. To address this, a new paragraph has been added to the end of the Introduction section, emphasizing the relevant concepts discussed. Here is the newly added paragraph:

“Finally, the Discussion addresses the comparison between the cytotoxicity of Ag and Au NPs against bacteria based on morphological characteristics, synthesis methods, and Gram typology. Additionally, the Conclusions, presented with thoroughness, summarize the importance of NPs in combating antibiotic resistance and propose future directions and biomedical applications, instilling a sense of confidence in our professional colleagues.”

Comment 14: This table is much more useful than the other ones. Consider to add some data (shape, synthesys method) to this table and to avoid the others  (Table 6).

Response 14: Additional data regarding the shape and synthesis method of the nanoparticles have been included in Table 6 (Table 4 in the new manuscript) to address the suggestion. This enhancement provides a more comprehensive overview of the factors influencing the antibacterial efficacy of Ag and Au NPs. Consequently, the other tables have been removed to ensure clarity and focus on the most relevant and helpful information.

Comment 15: Add the issue of the development of resistance to silver by bacteria. (On Chapter 8)

Response 15: A new section titled "10.2 Clinical Trials Involving Nanoparticles" has been added to describe various clinical trials and address the issue of bacterial resistance to Ag and Au NPs.

Round 2

Reviewer 1 Report

Comments and Suggestions for Authors

I am satisfied with the revised version. Thank you